# THE SPACETIME OF DIFFUSION MODELS: AN INFORMATION GEOMETRY PERSPECTIVE

**Rafał Karczewski[1], Markus Heinonen[1], Alison Pouplin[1], Søren Hauberg[2], Vikas Garg[1,3]**

[1] Aalto University, [2] Technical University of Denmark, [3] YaiYai Ltd

`{rafal.karczewski, markus.o.heinonen}@aalto.fi`
`alison.pouplin@gmail.com, sohau@dtu.dk, vgarg@csail.mit.edu`

## ABSTRACT

We present a novel geometric perspective on the latent space of diffusion models. We first show that the standard pullback approach, utilizing the deterministic probability flow ODE decoder, is fundamentally flawed. It provably forces geodesics to decode as straight segments in data space, effectively ignoring any intrinsic data geometry beyond the ambient Euclidean space. Complementing this view, diffusion also admits a stochastic decoder via the reverse SDE, which enables an information geometric treatment with the Fisher-Rao metric. However, a choice of $x_T$ as the latent representation collapses this metric due to memorylessness. We address this by introducing a latent spacetime $z = (x_t, t)$ that indexes the family of denoising distributions $p(x_0|x_t)$ across all noise scales, yielding a nontrivial geometric structure. We prove these distributions form an exponential family and derive simulation-free estimators for curve lengths, enabling efficient geodesic computation. The resulting structure induces a principled Diffusion Edit Distance, where geodesics trace minimal sequences of noise and denoise edits between data. We also demonstrate benefits for transition path sampling in molecular systems, including constrained variants such as low-variance transitions and region avoidance. Code is available at https://github.com/Aalto-QuML/spacetime-geometry.

## 1 INTRODUCTION

Diffusion models have emerged as a powerful paradigm for generative modeling, demonstrating remarkable success in learning to model and sample data (Yang et al., 2023). While the underlying mathematical frameworks of training and sampling are well-established (Sohl-Dickstein et al., 2015; Kingma et al., 2021; Song et al., 2021; Lu et al., 2022; Holderrieth et al., 2025), analysing how information evolves through the noisy intermediate states $x_t$ for $t \in [0, T]$ remains an open question. Our work addresses this by defining and analyzing the geometric structure of diffusion models, which provides a principled framework for understanding their inner workings.

In generative models, a common way to study the intrinsic geometry of the data is to pull back the ambient (Euclidean) metric onto the latent space (Arvanitidis et al., 2018; 2022). Equipped with this pullback metric, shortest paths (i.e., *geodesics*) in the latent space decode to realistic transitions along data that lie on a lower-dimensional submanifold.

In a diffusion model, a natural choice for the decoder is the reverse ODE $x_0(x_T)$, which allows us to derive the pullback geometry of the latents $x_T$. Interestingly, we prove that this leads to latent shortest paths always decoding to linear interpolations in data space, which have little practical utility.

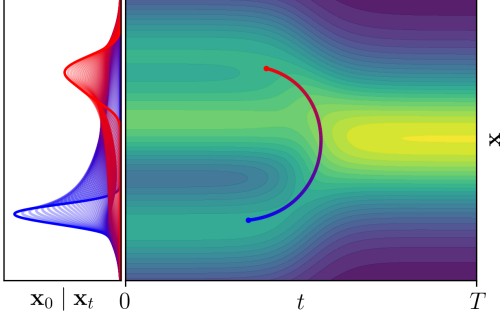

Figure 1: **A geodesic in spacetime** is the shortest path between denoising distributions.

We then turn our attention to the denoising posterior distribution $p(x_0|x_t)$ given by the reverse SDE. We propose an alternative *Fisher-Rao geometry*, which measures how the denoising distribution $p(x_0|x_t)$ changes when manipulating the latent $x_t$.

We introduce the Fisher-Rao metric $\mathbf{G}(\boldsymbol{x}_t, t)$ that varies with both state and time over the *latent spacetime* $(\boldsymbol{x}_t, t)$ (Fig. 1).

Estimating geodesics in information geometry is usually tractable only for analytic families. Although denoising distributions in diffusion are complex and non-Gaussian, we show that they form an exponential family. This simplifies the geometry and yields a practical method for computing geodesics between any two samples through the spacetime. In the Fisher–Rao setting, curve lengths can be evaluated without running the reverse SDE, which significantly reduces the computational cost.

We demonstrate the utility of the Fisher-Rao geometry in diffusion models in two ways. First, it induces a principled *Diffusion Edit Distance* on data that admits a clear interpretation: the geodesic between $\boldsymbol{x}^a$ and $\boldsymbol{x}^b$ traces the minimal sequence of edits, adding just enough noise to forget information specific to $\boldsymbol{x}^a$ and then denoising to introduce information specific to $\boldsymbol{x}^b$. The resulting length quantifies the total edit cost. Second, spacetime geodesics allow generating transition paths in molecular systems, where we obtain results competitive with specialized state-of-the-art methods and can incorporate constraints such as avoidance of designated regions in data space.

## 2 BACKGROUND ON DIFFUSION MODELS

We assume a data distribution $q$ defined on $\mathbb{R}^D$, and the forward process

$$p(\boldsymbol{x}_t|\boldsymbol{x}_0) = \mathcal{N}(\boldsymbol{x}_t|\alpha_t\boldsymbol{x}_0, \sigma_t^2\boldsymbol{I}), \tag{1}$$

which gradually transforms $q$ into pure noise $p_T \approx \mathcal{N}(\mathbf{0}, \sigma_T^2\boldsymbol{I})$ at time $T$, where $\alpha_t, \sigma_t$ define the forward drift $f_t$ and diffusion $g_t$. There exists a *denoising* SDE reverse process (Anderson, 1982)

$$\text{Reverse SDE:} \quad d\boldsymbol{x} = \left(f_t\boldsymbol{x} - g_t^2\nabla\log p_t(\boldsymbol{x})\right)dt + g_td\overline{\mathbf{W}}_t, \quad \boldsymbol{x}_T \sim p_T, \tag{2}$$

where $p_t$ is the marginal distribution of the forward process (Eq. 1) at time $t$, and $\overline{\mathbf{W}}$ is a reverse Wiener process. Somewhat unexpectedly, there exists a deterministic Probability Flow ODE (PF-ODE) with matching marginals (Song et al., 2021):

$$\text{PF ODE:} \quad d\boldsymbol{x} = \left(f_t\boldsymbol{x} - \frac{1}{2}g_t^2\nabla\log p_t(\boldsymbol{x})\right)dt, \quad \boldsymbol{x}_T \sim p_T. \tag{3}$$

Assuming we can approximate the score $\nabla\log p_t$ (Karras et al., 2024), we denote by $\boldsymbol{x}_T \mapsto \boldsymbol{x}_0(\boldsymbol{x}_T)$ the *deterministic* denoiser of solving the PF-ODE from noise $\boldsymbol{x}_T$, while we denote by $p(\boldsymbol{x}_0|\boldsymbol{x}_t)$ the denoising distributions induced by *stochastic* sampling of the reverse SDE (Karras et al., 2022).

## 3 RIEMANNIAN GEOMETRY OF DIFFUSION MODELS

Riemannian geometry equips a latent space $\mathcal{Z}$ with a smoothly varying *metric tensor* $\mathbf{G}(\boldsymbol{z})$ for $\boldsymbol{z} \in \mathcal{Z}$. This metric defines inner products and induces the notions of distance and curve length (Do Carmo & Francis, 1992). Several works have developed diffusion models *on top* of Riemannian manifolds, such as spheres, tori and hyperboloids (De Bortoli et al., 2022; Huang et al., 2022; Thornton et al., 2022). In this paper, we instead study what kind of Riemannian geometries are *implicitly induced* by the denoiser within a real vector space $\mathbb{R}^D$ (e.g., images).

In Euclidean geometry, the space is flat, with distances given by the length of straight lines connecting points. In Riemannian spaces, the shortest path between two points is no longer straight, but a curved *geodesic*. A smooth curve $\boldsymbol{\gamma} : [0, 1] \to \mathcal{Z}$ between fixed endpoints $\boldsymbol{\gamma}_0, \boldsymbol{\gamma}_1$ is a geodesic if it minimizes the length

$$\ell(\boldsymbol{\gamma}) = \int_0^1 \|\dot{\boldsymbol{\gamma}}_s\|_{\mathbf{G}}ds = \int_0^1 \sqrt{\dot{\boldsymbol{\gamma}}_s^T\mathbf{G}(\boldsymbol{\gamma}_s)\dot{\boldsymbol{\gamma}}_s}ds, \tag{4}$$

or, equivalently, the energy $\mathcal{E}(\boldsymbol{\gamma}) = \frac{1}{2}\int_0^1 \|\dot{\boldsymbol{\gamma}}_s\|_{\mathbf{G}}^2ds$.

We introduce two interpretations of Riemannian geometry $\mathbf{G}$ for diffusion models, depending on whether the decoder is *deterministic* or *stochastic*. In both cases, we first assume the latent space is the noise space $\boldsymbol{x}_T$, and later relax this to cover the entire noisy sample space $\boldsymbol{x}_t$.

**Deterministic sampler: pullback geometry.** Let $\boldsymbol{x}_T \mapsto \boldsymbol{x}_0(\boldsymbol{x}_T)$ be a deterministic map given by the PF-ODE (Eq. 3) mapping noise to data. We propose the pullback metric (Arvanitidis et al., 2022; Park et al., 2023)

$$\mathbf{G}_{\text{PB}}(\boldsymbol{x}_T) = \left(\frac{\partial \boldsymbol{x}_0}{\partial \boldsymbol{x}_T}\right)^\top \left(\frac{\partial \boldsymbol{x}_0}{\partial \boldsymbol{x}_T}\right) \in \mathbb{R}^{D \times D}, \qquad \boldsymbol{x}_0 := \boldsymbol{x}_0(\boldsymbol{x}_T) \in \mathbb{R}^D \tag{5}$$

which measures how an *infinitesimal noise step* $d\boldsymbol{x}_T$ changes the decoded sample:

$$\left\|\boldsymbol{x}_0(\boldsymbol{x}_T + d\boldsymbol{x}_T) - \boldsymbol{x}_0(\boldsymbol{x}_T)\right\|^2 = d\boldsymbol{x}_T^\top \mathbf{G}_{\text{PB}}(\boldsymbol{x}_T) d\boldsymbol{x}_T + o(\|d\boldsymbol{x}_T\|^2). \tag{6}$$

**Stochastic sampler: information geometry.** Alternatively, consider a *stochastic* decoder that, for each latent $\boldsymbol{x}_T$ defines a denoising distribution $p(\boldsymbol{x}_0|\boldsymbol{x}_T)$ by solving the Reverse SDE (Eq. 2). We propose the information-geometric viewpoint via the Fisher-Rao metric (Amari, 2016)

$$\mathbf{G}_{\text{IG}}(\boldsymbol{x}_T) = \mathbb{E}_{\boldsymbol{x}_0 \sim p(\boldsymbol{x}_0|\boldsymbol{x}_T)} \left[\nabla_{\boldsymbol{x}_T} \log p(\boldsymbol{x}_0|\boldsymbol{x}_T) \nabla_{\boldsymbol{x}_T} \log p(\boldsymbol{x}_0|\boldsymbol{x}_T)^\top\right] \in \mathbb{R}^{D \times D}, \tag{7}$$

which measures how an *infinitesimal noise step* $d\boldsymbol{x}_T$ changes the *entire* denoising distribution:

$$\text{KL}\left[p(\boldsymbol{x}_0 \mid \boldsymbol{x}_T) \, \| \, p(\boldsymbol{x}_0 \mid \boldsymbol{x}_T + d\boldsymbol{x}_T)\right] = \frac{1}{2} d\boldsymbol{x}_T^\top \mathbf{G}_{\text{IG}}(\boldsymbol{x}_T) d\boldsymbol{x}_T + o(\|d\boldsymbol{x}_T\|^2). \tag{8}$$

For a helpful tutorial on information geometry, we refer to Mishra et al. (2023).

## 4 PULLBACK GEOMETRY COLLAPSES IN DIFFUSION MODELS

Both pullback and information geometries are, in principle, applicable. We will first show the pullback geometry has fundamental theoretical limitations in diffusion models, rendering it practically useless.

Assume we estimate a geodesic $\boldsymbol{\gamma}$ in the noise space $\boldsymbol{x}_T$ such that its endpoints decode to $\boldsymbol{x}_0(\boldsymbol{\gamma}_0) = \boldsymbol{x}^a$ and $\boldsymbol{x}_0(\boldsymbol{\gamma}_1) = \boldsymbol{x}^b$. The pullback energy $\mathcal{E}(\boldsymbol{\gamma})$ (Eq. 4) can be shown to only depend on the decoded curve $\boldsymbol{x}_0(\boldsymbol{\gamma}_s)$ in data space (See Appendix B):

$$\mathcal{E}_{\text{PB}}(\boldsymbol{\gamma}) = \frac{1}{2} \int_0^1 \left\|\frac{d}{ds} \boldsymbol{x}_0(\boldsymbol{\gamma}_s)\right\|^2 ds. \tag{9}$$

The unique minimizer is the constant-speed straight line $\boldsymbol{x}_s = (1-s)\boldsymbol{x}^a + s\boldsymbol{x}^b$ in data space. Since the ODE is bijective, this line has a unique latent preimage $\boldsymbol{\gamma}_s^\star = \boldsymbol{x}_0^{-1}(\boldsymbol{x}_s) = \boldsymbol{x}_T(\boldsymbol{x}_s)$, which is thus a pullback geodesic, and the energy reduces to Euclidean distance in data space:

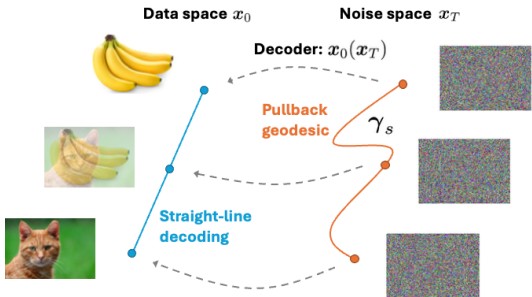

Figure 2: **The pullback geodesics curve in noise space, but decode to straight lines in data space.**

$$\mathcal{E}_{\text{PB}}(\boldsymbol{\gamma}) := \frac{1}{2}\left\|\boldsymbol{x}^a - \boldsymbol{x}^b\right\|^2. \tag{10}$$

Hence, *all* pullback geodesics decode to straight segments, ignoring the curvature of the data manifold and undermining downstream applications (See Fig. 2). The same pathology applies for denoised geodesics in the intermediate space $\boldsymbol{x}_t$ as well. The core reason for this is that, in diffusion models, the latent and data spaces have the same dimension. The decoder operates directly in the ambient space and, without further dimensional constraints, it cannot capture the intrinsic structure of the data, even if the data lie on a lower-dimensional submanifold. As a result, the standard pullback metric provides no meaningful geometric information. A formal proof and discussion are in Appendix B.

## 5 INFORMATION GEOMETRY WITH DENOISING DECODERS

Under the stochastic view, the decoder is the denoising distribution $p(\boldsymbol{x}_0|\boldsymbol{x}_T)$ obtained by reversing the diffusion process (Eq. 2). This yields a family of distributions on the data space parametrized with noise vectors $\boldsymbol{x}_T$. The information geometry assigns the Fisher-Rao metric to the latent domain, and geodesic energies/lengths are computed as in Section 3.

**The latent spacetime.** Diffusion models are "memoryless" (Domingo-Enrich et al., 2025):

$$p(\boldsymbol{x}_T \mid \boldsymbol{x}_0) \approx p_T(\boldsymbol{x}_T) \quad \Rightarrow \quad p(\boldsymbol{x}_0 \mid \boldsymbol{x}_T) \approx q(\boldsymbol{x}_0). \tag{11}$$

Hence $p(\boldsymbol{x}_0 \mid \boldsymbol{x}_T)$ is (approximately) independent of $\boldsymbol{x}_T$, implying $\nabla_{\boldsymbol{x}_T} \log p(\boldsymbol{x}_0 \mid \boldsymbol{x}_T) \approx 0$ and a collapse of the Fisher–Rao metric, $\mathbf{G}_{\text{IG}} \approx \mathbf{0}$ (Eq. 7). Consequently, if we identify the latent space with $\boldsymbol{z} = \boldsymbol{x}_T$, all $\boldsymbol{x}_T$ become metrically indistinguishable. This could be avoided by choosing $\boldsymbol{z} = \boldsymbol{x}_t$ for some $t < T$; however, instead of choosing an arbitrary noise level $t$, we propose to model all noise levels simultaneously by considering points in the $(D+1)$-dimensional latent *spacetime*

$$\boldsymbol{z} = (\boldsymbol{x}_t, t) \in \mathbb{R}^D \times (0, T], \tag{12}$$

which define the family of all denoising distributions $\{p(\boldsymbol{x}_0 | \boldsymbol{x}_t)\}$ across all noise levels (Fig. 1).

**Why include time?** The resulting Fisher-Rao metric $\mathbf{G}_{\text{IG}}(\boldsymbol{z})$ varies with state and time, restoring a nontrivial geometry and enabling navigation across noise levels within a unified structure. Identifying clean data with spacetime points $(\boldsymbol{x}, 0)$, for which $p(\boldsymbol{x}_0 \mid \boldsymbol{x}_0 = \boldsymbol{x}) = \delta_{\boldsymbol{x}}$, lets geodesics *connect clean endpoints through noisy intermediates*. This yields (i) a principled notion of distance between data as the length of the shortest spacetime path (Diffusion Edit Distance), and (ii) a mechanism for transition-path sampling via spacetime geodesics; both are demonstrated empirically in Section 6.

**Tractable energy estimation.** Usually, the information-geometric energy of a discretized curve $\boldsymbol{\gamma} = \{\boldsymbol{z}_n\}_{n=0}^{N-1}$ is approximated via the local-KL approximation (Arvanitidis et al., 2022):

$$\mathcal{E}(\boldsymbol{\gamma}) \approx (N-1) \sum_{n=0}^{N-2} \text{KL}\Big[p(\cdot \mid \boldsymbol{z}_n) \,\big\|\, p(\cdot \mid \boldsymbol{z}_{n+1})\Big], \tag{13}$$

but such KLs are generally intractable, unless $p(\cdot|\boldsymbol{z})$ is a simple analytic distribution such as multinomial or Gaussian, which is not the case for denoising distributions $p(\boldsymbol{x}_0|\boldsymbol{x}_t)$. Nonetheless, we show that in the specific case of the diffusion spacetime, the energy can be tractably estimated.

**Proposition 5.1** (Spacetime energy estimation - informal). *The energy of discretized spacetime curve* $\boldsymbol{\gamma} = \{\boldsymbol{z}_n\}_{n=0}^{N-1}$ *with* $\boldsymbol{z}_n = (\boldsymbol{x}_{t_n}, t_n)$ *admits an approximation*

$$\mathcal{E}(\boldsymbol{\gamma}) \approx \frac{N-1}{2} \sum_{n=0}^{N-2} \Big(\boldsymbol{\eta}(\boldsymbol{z}_{n+1}) - \boldsymbol{\eta}(\boldsymbol{z}_n)\Big)^\top \Big(\boldsymbol{\mu}(\boldsymbol{z}_{n+1}) - \boldsymbol{\mu}(\boldsymbol{z}_n)\Big), \tag{14}$$

*where*

$$\boldsymbol{\eta}(\boldsymbol{x}_t, t) = \left(\frac{\alpha_t}{\sigma_t^2}\boldsymbol{x}_t, \; -\frac{\alpha_t^2}{2\sigma_t^2}\right), \qquad \boldsymbol{\mu}(\boldsymbol{x}_t, t) = \left(\mathbb{E}[\boldsymbol{x}_0 \mid \boldsymbol{x}_t], \; \mathbb{E}[\|\boldsymbol{x}_0\|^2 \mid \boldsymbol{x}_t]\right). \tag{15}$$

The proof (Appendix C) consists of showing that denoising distributions form an exponential family, which admits a simplified energy formula. In practice, we calculate $\boldsymbol{\mu}(\boldsymbol{x}_t, t)$ with Tweedie's formula over the approximate denoiser $\hat{\boldsymbol{x}}_0(\boldsymbol{x}_t)$ (See Appendix C.2 for details),

$$\begin{aligned}
\mathbb{E}[\boldsymbol{x}_0 \mid \boldsymbol{x}_t] &\approx \hat{\boldsymbol{x}}_0(\boldsymbol{x}_t) \\
\mathbb{E}[\|\boldsymbol{x}_0\|^2 \mid \boldsymbol{x}_t] &\approx \|\hat{\boldsymbol{x}}_0(\boldsymbol{x}_t)\|^2 + \frac{\sigma_t^2}{\alpha_t} \text{div}_{\boldsymbol{x}_t} \hat{\boldsymbol{x}}_0(\boldsymbol{x}_t),
\end{aligned} \tag{16}$$

where both $\hat{\boldsymbol{x}}_0$ and $\text{div}\,\hat{\boldsymbol{x}}_0$ are computed efficiently via Hutchinson's trick (Hutchinson, 1989; Grathwohl et al., 2019), enabling the esimation of $\boldsymbol{\mu}(\boldsymbol{x}_t, t)$ with a single Jacobian-vector product (JVP).

> Spacetime geodesics are simulation-free: the energy calculation requires only $N$ JVPs of the denoiser $\hat{\boldsymbol{x}}_0$ for a curve discretized into $N$ points.

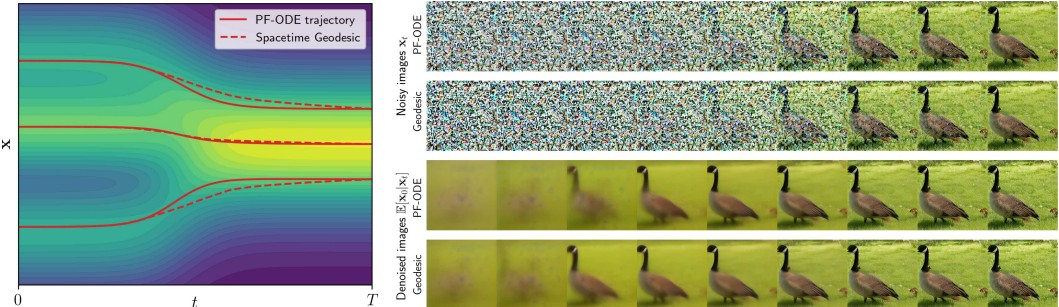

Figure 3: **PF-ODE paths are similar to energy-minimizing geodesics**. Left: Geodesics move in straighter lines than PF-ODE trajectories in 1D toy density. Right: Geodesics are almost indistinguishable to PF-ODE sampling in ImageNet-512 EDM2 model.

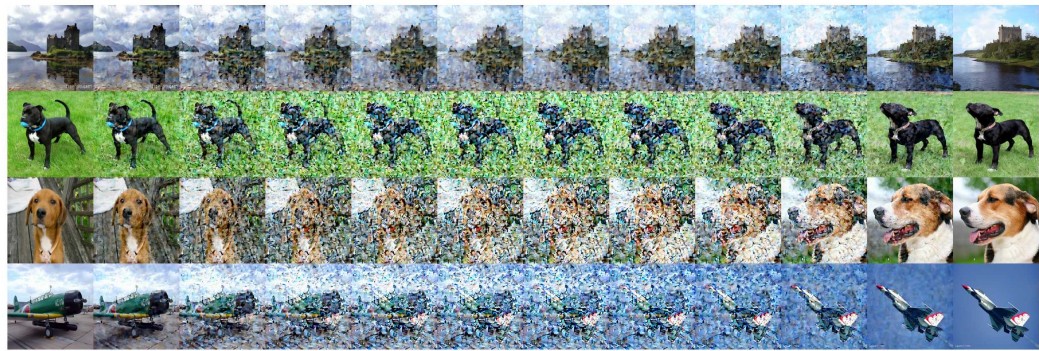

Figure 4: **Spacetime geodesics between images.** Each row shows a geodesic $\gamma$ between clean images. The path passes through noisy states and then denoises, realizing the minimal total edit between endpoints. Its length $\ell(\gamma)$ is the Diffusion Edit Distance (DiffED), which measures how much the denoising distribution changes along the optimal traversal.

## 6 EXPERIMENTS

### 6.1 SAMPLING TRAJECTORIES

We compare the trajectories obtained by solving the PF-ODE $\boldsymbol{x}_0(\boldsymbol{x}_T)$ (Eq. 3) with geodesics between the same endpoints $\boldsymbol{x}_0, \boldsymbol{x}_T$. For a toy example of 1D mixture of Gaussians, we observe the geodesics curving less than the PF-ODE trajectories in the early sampling (high $t$), while being indistinguishable for lower values of $t$ (See Fig. 3 left and Appendix G.1 for details).

We find only marginal perceptual difference between the PF-ODE sampling trajectories and the geodesics in the EDM2 ImageNet-512 model (Karras et al., 2024). The geodesic appears to generate information slightly earlier, but the difference is minor (See Fig. 3 right, and Appendix G.2 for details).

We note that spacetime geodesics are not an alternative sampling method since they require knowing the endpoints beforehand. An investigation into whether our framework can be used to improve sampling strategies is an interesting future research direction.

### 6.2 DIFFUSION EDIT DISTANCE

The spacetime geometry yields a principled distance on the data space. We identify clean datum $\boldsymbol{x} \in \mathbb{R}^d$ with the spacetime point $(\boldsymbol{x}, 0)$, corresponding to the Dirac denoising distribution $\delta_{\boldsymbol{x}}$. Given two points $\boldsymbol{x}^a, \boldsymbol{x}^b$, we define the *Diffusion Edit Distance* (DiffED) by

$$\text{DiffED}(\boldsymbol{x}^a, \boldsymbol{x}^b) = \ell(\boldsymbol{\gamma}), \tag{17}$$

where $\boldsymbol{\gamma}$ is the spacetime geodesic between $(\boldsymbol{x}^a, 0)$ and $(\boldsymbol{x}^b, 0)$. For numerical stability, we anchor endpoints at a small $t_{\min} > 0$ rather than at 0. See Algorithm 4 for DiffED pseudocode.

A spacetime geodesic links two clean data points through intermediate noisy states. It can be interpreted as the minimal sequence of edits: add just enough noise to discard information specific to $\boldsymbol{x}^a$, then remove noise to introduce information specific to $\boldsymbol{x}^b$. The path length is the total edit cost, which is measured by how much the denoising distribution changes along the path. Fig. 4 visualizes the spacetime geodesics: as endpoint similarity decreases, the intermediate points become noisier.

We quantitatively evaluate DiffED on image data. First, we ask whether DiffED correlates with human perception as approximated by Learned Perceptual Image Patch Similarity (LPIPS) (Zhang et al., 2018). We randomly selected 10 classes in the ImageNet dataset and sampled 20 random image pairs for each. We then evaluated the DiffED and LPIPS for each image pair, and found the correlation to be very low at approximately -7%, suggesting that perceptual similarity and geometric edit cost capture different notions of closeness. We found DiffED to be more closely related to the structural similarity index measure (SSIM) (Wang et al., 2004), which correlates at 53% with DiffED.

To qualitatively compare different notions of image similarity, we order image pairs by their similarity evaluated with multiple metrics: DiffED, LPIPS, SSIM, and Euclidean. We show the results in Fig. 8.

## 6.3 TRANSITION PATH SAMPLING

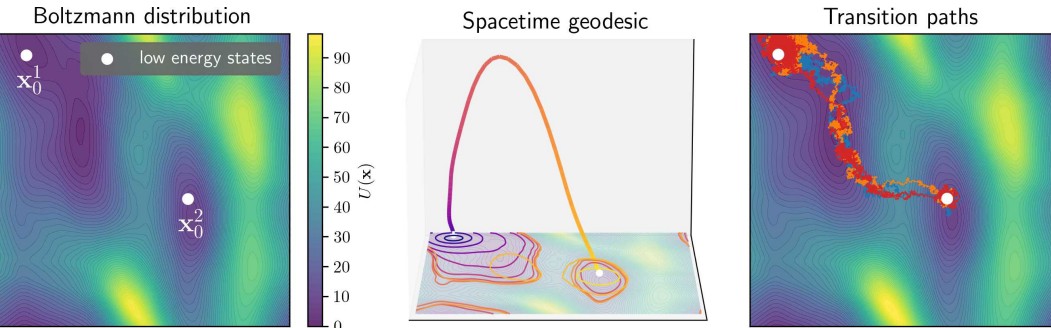

Figure 5: **Spacetime geodesics enable sampling transition paths between low-energy states.** Left: Alanine Dipeptide energy landscape wrt two dihedral angles, with two energy minima $\boldsymbol{x}_0^1, \boldsymbol{x}_0^2$. Middle: Spacetime geodesic $\boldsymbol{\gamma}$ connecting $\boldsymbol{x}_0^1$ and $\boldsymbol{x}_0^2$. Right: Annealed Langevin transition path samples.

Another application of the spacetime geometry is the problem of transition-path sampling (Holdijk et al., 2023; Du et al., 2024; Raja et al., 2025), whose goal is to find probable transition paths between low-energy states. We assume a Boltzmann distribution

$$q(\boldsymbol{x}) \propto \exp(-U(\boldsymbol{x})), \tag{18}$$

where $U$ is a known energy function, which is a common assumption in molecular dynamics. In this setting, the denoising distribution follows a tractable energy function (See Eq. 60)

$$p(\boldsymbol{x}_0|\boldsymbol{x}_t) \propto q(\boldsymbol{x}_0)p(\boldsymbol{x}_t|\boldsymbol{x}_0) \propto \exp\left(\underbrace{-U(\boldsymbol{x}_0) - \tfrac{1}{2}\mathrm{SNR}(t)\left\|\boldsymbol{x}_0 - \boldsymbol{x}_t/\alpha_t\right\|^2}_{-U(\boldsymbol{x}_0|\boldsymbol{x}_t)}\right). \tag{19}$$

To construct a transition path between two low-energy states $\boldsymbol{x}_0^1$ and $\boldsymbol{x}_0^2$, we estimate the spacetime geodesic $\boldsymbol{\gamma}$ between them using a denoiser model $\hat{\boldsymbol{x}}_0(\boldsymbol{x}_t) \approx \mathbb{E}[\boldsymbol{x}_0|\boldsymbol{x}_t]$ with Proposition 5.1, as shown in Fig. 5. At each interpolation point $s \in [0,1]$, the geodesic defines a denoising Boltzmann distribution $p(\boldsymbol{x}|\boldsymbol{\gamma}_s)$ where $U(\boldsymbol{x}|\boldsymbol{\gamma}_s)$ is the energy at that spacetime location. See Appendix G.3 for details.

**Annealed Langevin Dynamics.** To sample transition paths, we use Langevin dynamics

$$d\boldsymbol{x} = -\nabla_{\boldsymbol{x}}U(\boldsymbol{x}|\boldsymbol{\gamma}_s)dt + \sqrt{2}d\mathrm{W}_t, \tag{20}$$

whose stationary distributions are $p(\boldsymbol{x} \mid \boldsymbol{\gamma}_s) \propto \exp(-U(\boldsymbol{x}|\boldsymbol{\gamma}_s))$ for any $s$. To obtain the trajectories from $\boldsymbol{x}_0^1$ to $\boldsymbol{x}_0^2$, we gradually increase $s$ from 0 to 1 using annealed Langevin (Song & Ermon, 2019).

| MCMC variable length | MCMC fixed length | Doob's lagrangian | Spacetime geodesic + ALD |
|---|---|---|---|
| 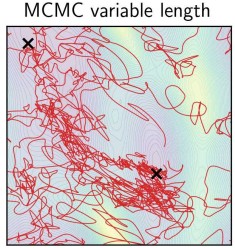 | 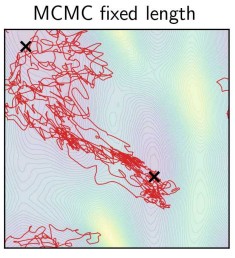 | 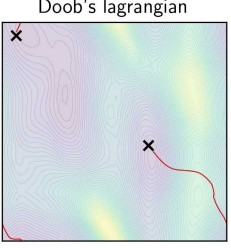 | 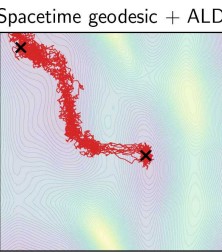 |

Figure 6: **Transition paths generated with a spacetime geodesic avoid high-energy regions without collapsing to a single path.** Compared with MCMC baselines, the spacetime-geodesic method yields transition paths that better avoid high-energy areas, whereas Doob's Lagrangian collapses to generating nearly identical trajectories. Ten sample paths are shown for each method.

After discretizing the geodesic into $N$ points $\boldsymbol{\gamma}_n$, we alternate between taking $K$ steps of Eq. 20 conditioned on $\boldsymbol{\gamma}_n$ and updating $\boldsymbol{\gamma}_n \mapsto \boldsymbol{\gamma}_{n+1}$, as described in Algorithm 1. This approach assumes that $p(\boldsymbol{x}|\boldsymbol{\gamma}_n)$ is *close* to $p(\boldsymbol{x}|\boldsymbol{\gamma}_{n+1})$, and thus $\boldsymbol{x} \sim p(\boldsymbol{x}|\boldsymbol{\gamma}_n)$ is a good starting point to Langevin dynamics conditioned on $\boldsymbol{\gamma}_{n+1}$.

Table 1: **Spacetime geodesics outperform methods tailored to transition path sampling**. Parentheses denote extra energy evaluations used to generate training data for the base diffusion model, which do not scale with the number of generated paths. Baseline details in Appendix H.

| | **MaxEnergy** ($\downarrow$) | **# Evaluations** ($\downarrow$) |
|---|---|---|
| Lower Bound | 36.42 | N/A |
| MCMC-fixed-length | $42.54 \pm 7.42$ | 1.29B |
| MCMC-variable-length | $58.11 \pm 18.51$ | 21.02M |
| Doob's Lagrangian (Du et al., 2024) | $66.24 \pm 1.01$ | 38.4M |
| Spacetime geodesic (Ours) | $\mathbf{37.36} \pm 0.60$ | 16M (+16M) |

**Alanine dipeptide.** We compute a spacetime geodesic connecting two molecular configurations of Alanine Dipeptide, as in Holdijk et al. (2023). In Fig. 5, the energy landscape is visualized over the dihedral angle space, with a neural network used to approximate the potential energy $U$. Using our trained denoiser $\hat{\boldsymbol{x}}_0(\boldsymbol{x}_t)$, we estimate the expectation parameter $\boldsymbol{\mu}$, which allows us to compute and visualize a geodesic trajectory through spacetime. Transition paths were generated using Algorithm 1. See Appendix G.3 for details.

**Baselines.** We considered Holdijk et al. (2023); Du et al. (2024); Raja et al. (2025) and adopt Doob's Lagrangian (Du et al., 2024); the others were excluded due to reproducibility issues (see Appendix H). We also evaluate two MCMC two-way shooting variants (Brotzakis & Bolhuis, 2016)-uniform point selection with variable or fixed trajectory length-using transition paths from the official Du et al. (2024) code release. For each method we generate 1,000 paths and report mean MaxEnergy (lower is better) and its numerical lower bound $\min_{\boldsymbol{\gamma}} \max_s U(\boldsymbol{\gamma}_s)$, along with the number of energy evaluations needed for 1,000 paths. To train a base diffusion model for our method, we generated data using Langevin dynamics (16M[1] energy evaluations), a one-time cost that does not scale with the number of generated transition paths.

**Results.** We show in Table 1 that our method outperforms the baselines in the MaxEnergy obtained along the transition paths. It is also considerably closer to the lower bound than to the next best baseline (MCMC-fixed length) while requiring several orders of magnitude fewer energy function evaluations. In Fig. 6, we show a qualitative comparison of transition paths generated with our method and the baselines. Our proposed method shows improved efficiency in avoiding high-energy

---

[1]+16M is the number of energy function evaluations to generate the training set with Langevin dynamics for the base diffusion model. We did not tune this number, and fewer evaluations may yield comparable performance.

regions compared to MCMC. In contrast, the Doob's Lagrangian method converged to a suboptimal solution, producing nearly identical transition paths. We discuss this in more detail in Appendix H.

---

**Algorithm 1** Transition Path Sampling with Annealed Langevin Dynamics

---

**Require:** $\boldsymbol{x}_a, \boldsymbol{x}_b \in \mathbb{R}^D$ endpoints, $N_{\boldsymbol{\gamma}} > 0$, $T > 0$, $t_{\min}$, $dt$
1: $\boldsymbol{\gamma} \leftarrow \text{SPACETIMEGEODESIC}(\boldsymbol{x}_a, \boldsymbol{x}_b)$       $\triangleright$ Approximate spacetime geodesic with Algorithm 3
2: $\mathcal{T} \leftarrow \{\boldsymbol{x} := \boldsymbol{x}_a\}$       $\triangleright$ Initialize chain $\mathcal{T}$ at $\boldsymbol{x}_a$
3: **for** $n \in \{0, \ldots, N_{\boldsymbol{\gamma}} - 1\}$ **do**       $\triangleright$ Iterate over the points on the geodesic $\boldsymbol{\gamma}_n$
4:      **for** $t \in \{1, \ldots, T\}$ **do**
5:          $\boldsymbol{\varepsilon} \sim \mathcal{N}(\boldsymbol{0}, \boldsymbol{I})$       $\triangleright$ Sample Gaussian noise
6:          $\boldsymbol{x} \leftarrow \boldsymbol{x} - \nabla_{\boldsymbol{x}} U(\boldsymbol{x}|\boldsymbol{\gamma}_n) dt + \sqrt{2dt}\boldsymbol{\varepsilon}$       $\triangleright$ Langevin update
7:          $\mathcal{T} \leftarrow \mathcal{T} \cup \{\boldsymbol{x}\}$       $\triangleright$ Append state $\boldsymbol{x}$ to chain
8:      **end for**
9: **end for**
10: **return** $\mathcal{T}$       $\triangleright$ Return chain

---

### 6.4 CONSTRAINED PATH SAMPLING

Suppose we would like to impose additional constraints along the geodesic interpolants. This corresponds to penalized optimization (Rygaard et al. (2025) also explore regularized geodesics)

$$\min_{\boldsymbol{\gamma}} \left\{ \mathcal{E}(\boldsymbol{\gamma}) + \lambda \int_0^1 h(\boldsymbol{\gamma}_s) ds, \quad \text{s.t.} \quad \boldsymbol{\gamma}_0 = (\boldsymbol{x}_0^1, 0), \boldsymbol{\gamma}_1 = (\boldsymbol{x}_0^2, 0) \right\}, \tag{21}$$

where $h : \mathbb{R} \times \mathbb{R}^D \to \mathbb{R}$ is some penalty function with $\lambda > 0$. We demonstrate the principle by (i) penalizing transition path variance, and (ii) imposing regions to avoid in the data space..

**Low-variance transitions.** Suppose we want the posterior $p(\boldsymbol{x} \mid \boldsymbol{\gamma}_s)$ to have a low variance. This concentrates the path around a narrower set of plausible states, more repeatable trajectories, albeit at the cost of reduced coverage. By Eq. 56, higher $\text{SNR}(t)$ yields lower denoising variance, so we implement this by penalizing low SNR via $h(\boldsymbol{x}, t) = \max(-\log \text{SNR}(t), \rho)$ for some threshold $\rho$.

**Avoiding restricted regions.** Suppose we want to avoid certain regions in the data space in the transition paths. We encode the region to avoid as a denoising distribution $p(\cdot|\boldsymbol{z}^*)$ for some $\boldsymbol{z}^* = (\boldsymbol{x}_t^*, t^*)$ where larger the $t^*$, larger the restricted region. We encode the penalty as KL distance between the denoising distributions (See Appendix D for the derivation)

$$\text{KL}\left[p(\cdot|\boldsymbol{z}^*)||p(\cdot|\boldsymbol{\gamma}_s)\right] = \int_0^s \left(\tfrac{d}{du}\boldsymbol{\eta}(\boldsymbol{\gamma}_u)\right)^\top (\boldsymbol{\mu}(\boldsymbol{\gamma}_u) - \boldsymbol{\mu}(\boldsymbol{z}^*)) \, du + C \tag{22}$$

$$h(\boldsymbol{\gamma}_s) = \min\left(\rho, -\text{KL}\left[p(\cdot|\boldsymbol{z}^*)||p(\cdot|\boldsymbol{\gamma}_s)\right]\right). \tag{23}$$

In Fig. 7, we compare spacetime geodesics (unconstrained) with low-variance, and region-avoiding spacetime curves. We visualize both the curves and the corresponding transition paths generated with Algorithm 1. This demonstrates that our framework with the penalized optimization (Eq. 21) can incorporate various preferences on the transition paths.

## 7 RELATED WORKS

We review three directions of research related to ours: (i) studies of latent noise in diffusion models, (ii) applications of information geometry in generative modeling, and (iii) geometric formulations for sampling efficiency.

**Latent–data geometry.** Several works analyze the relation between latent noise $\boldsymbol{x}_t$ and data $\boldsymbol{x}_0$. Yu et al. (2025) define a geodesic density in diffusion latent space; Park et al. (2023) apply Riemannian geometry to lower-dimensional latent codes; Karczewski et al. (2025) study how noise scaling affects log-densities and perceptual detail. Our work also investigates the $\boldsymbol{x}_t$ to $\boldsymbol{x}_0$ relationship but (a) uses the Fisher–Rao metric rather than an inverse-density metric, (b) retains the full-dimensional latent space without projection, and (c) analyzes the complete diffusion path across all timesteps.

Figure 7: **Vanilla transition paths can be constrained to have lower variance, or successfully avoid a restricted region** $p(\cdot|\boldsymbol{z}^*)$. Left: geodesics $\boldsymbol{\gamma}$. Right: transition paths $\mathcal{T}$.

**Information geometry in generative models.** Lobashev et al. (2025) introduce the Fisher–Rao metric on families $p(\boldsymbol{x}|\theta)$ to study phase-like transitions, where $\theta$ is a low-dimensional variable parametrizing a microstate $\boldsymbol{x}$. In contrast, we place the geometry on diffusion's explicit spacetime coordinates $\boldsymbol{z} = (\boldsymbol{x}_t, t)$, induced by the denoising posterior $p(\boldsymbol{x}_0|\boldsymbol{x}_t)$.

**Geometric approaches to sampling.** Two recent works also formulate diffusion models geometrically to improve sampling efficiency. Das et al. (2023) optimize the forward noising process by following the shortest geodesic between $p_0$ and $p_t$ under the Fisher-Rao metric, assuming $p_0(\boldsymbol{x}_0)$ to be Gaussian. Ghimire et al. (2023) model both the forward and reverse processes as Wasserstein gradient flows. Our contribution differs: we use information geometry (not optimal transport), focus on the reverse process (not the forward), and only require $p_0$ to admit a density.

## 8 LIMITATIONS

Although our framework defines geodesics between any noisy samples, optimizing between nearly clean ones is numerically unstable because their denoising distributions collapse to Dirac deltas, making Fisher-Rao (via local KL) distances effectively infinite. Therefore, consistent with diffusion practice (Song et al., 2021; Lu et al., 2022), we choose endpoints with non-negligible noise for tractable optimization (details in Appendix G).

The proposed distance metric DiffED (Section 6.2) is considerably slower (details in Appendix G.2) than established image similarity metrics such as LPIPS (Zhang et al., 2018), or SSIM (Wang et al., 2004). Exploring a distillation strategy involving training a separate model trained to predict DiffED is a possible future research direction.

## 9 CONCLUSION

We proposed a novel perspective on the latent space of diffusion models by viewing it as a $(D + 1)$-dimensional statistical manifold, with the Fisher-Rao metric inducing a geometrical structure. By leveraging the fact that the denoising distributions form an exponential family, we showed that we can tractably estimate geodesics even for high-dimensional image diffusion models. We visualized our methods for image interpolations and demonstrated their utility in molecular transition path sampling.

This work deepens our understanding of the latent space in diffusion models and has the potential to inspire further research, including the development of novel applications of the spacetime geometric framework, such as enhanced sampling techniques.

REPRODUCIBILITY STATEMENT

We include the source code in our submission, which allows for reproducing the results. Our claims made in the main text are proven in the appendices. Experiment details can be found in Appendix G.

ETHICS STATEMENT

The use of generative models, especially those capable of producing images and videos, poses considerable risks for misuse. Such technologies have the potential to produce harmful societal

effects, primarily through the spread of disinformation, but also by reinforcing harmful stereotypes and implicit biases. In this work, we contribute to a deeper understanding of diffusion models, which currently represent the leading methodology in generative modeling. While this insight may eventually support improvements to these models, thereby increasing the risk of misuse, it is important to note that our research does not introduce any new ethical risks beyond those already associated with generative AI.

We have used Large Language Models to polish writing on a sentence level.

## ACKNOWLEDGMENTS

This work was supported by the Finnish Center for Artificial Intelligence (FCAI) under Flagship R5 (award 15011052). SH was supported by research grants from VILLUM FONDEN (42062), the Novo Nordisk Foundation through the Center for Basic Research in Life Science (NNF20OC0062606), and the European Research Council (ERC) under the European Union's Horizon Programme (grant agreement 101125003). VG acknowledges the support from Saab-WASP (grant 411025), Academy of Finland (grant 342077), and the Jane and Aatos Erkko Foundation (grant 7001703).

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

## A    NOTATION

We denote $\boldsymbol{x} = (x^1, \ldots, x^D)^\top \in \mathbb{R}^D$ a point in $D$-dimensional Euclidean space (a column vector), $\mathrm{Tr}(\boldsymbol{A}) = \sum_i A_{ii}$ - the trace operator of a square matrix $\boldsymbol{A} \in \mathbb{R}^{k \times k}$.

**Differential operators.**    For a scalar function $f : \mathbb{R}^D \to \mathbb{R}, \boldsymbol{x} \mapsto f(\boldsymbol{x}) \in \mathbb{R}$, we denote

$$\text{gradient:} \quad \nabla_{\boldsymbol{x}} f(\tilde{\boldsymbol{x}}) = \left( \frac{\partial f}{\partial x^1}, \ldots, \frac{\partial f}{\partial x^D} \right)^\top \Bigg|_{\boldsymbol{x} = \tilde{\boldsymbol{x}}} \qquad \in \mathbb{R}^D$$

$$\text{Hessian:} \quad \nabla_{\boldsymbol{x}}^2 f(\tilde{\boldsymbol{x}}) = \left[ \frac{\partial^2 f}{\partial x^i \partial x^j} \right]_{i,j} \Bigg|_{\boldsymbol{x} = \tilde{\boldsymbol{x}}} \qquad \in \mathbb{R}^{D \times D}$$

$$\text{Laplacian:} \quad \Delta_{\boldsymbol{x}} f(\tilde{\boldsymbol{x}}) = \mathrm{Tr}\left( \nabla_{\boldsymbol{x}}^2 f(\tilde{\boldsymbol{x}}) \right) = \sum_{i=1}^D \frac{\partial^2 f}{\partial (x^i)^2} \Bigg|_{\boldsymbol{x} = \tilde{\boldsymbol{x}}} \qquad \in \mathbb{R}.$$

For a curve $\boldsymbol{\gamma} : [0, 1] \to \mathbb{R}^k, s \mapsto \boldsymbol{\gamma}_s \in \mathbb{R}^k$ we denote

$$\text{time derivative:} \quad \dot{\boldsymbol{\gamma}}_s = \frac{d}{ds} \boldsymbol{\gamma}_s \in \mathbb{R}^k.$$

For a vector valued function $f : \mathbb{R}^k \to \mathbb{R}^m, \boldsymbol{x} \mapsto (f^1(\boldsymbol{x}), \ldots, f^m(\boldsymbol{x}))^\top \in \mathbb{R}^m$ we denote

$$\text{Jacobian:} \quad \frac{\partial f(\tilde{\boldsymbol{x}})}{\partial \boldsymbol{x}} = \left[ \frac{\partial f^i}{\partial x^j} \right]_{i,j} \Bigg|_{\boldsymbol{x} = \tilde{\boldsymbol{x}}} \in \mathbb{R}^{m \times k}$$

When $k = m$, we define

$$\text{divergence:} \quad \mathrm{div}_{\boldsymbol{x}} f(\tilde{\boldsymbol{x}}) = \mathrm{Tr}\left( \frac{\partial f(\tilde{\boldsymbol{x}})}{\partial \boldsymbol{x}} \right) = \sum_{i=1}^k \frac{\partial f^i}{\partial x^i} \Bigg|_{\boldsymbol{x} = \tilde{\boldsymbol{x}}} \in \mathbb{R}$$

**Functions with two arguments.**    For $f : \mathbb{R}^{k_1} \times \mathbb{R}^{k_2} \to \mathbb{R}, (\boldsymbol{x}_1, \boldsymbol{x}_2) \mapsto f(\boldsymbol{x}_1, \boldsymbol{x}_2) \in \mathbb{R}$ we define (analogously w.r.t. second argument)

$$\text{gradient w.r.t. first argument:} \quad \nabla_{\boldsymbol{x}_1} f(\tilde{\boldsymbol{x}}_1, \tilde{\boldsymbol{x}}_2) = \left( \frac{\partial f}{\partial x_1^1}, \ldots, \frac{\partial f}{\partial x_1^{k_1}} \right)^\top \Bigg|_{(\boldsymbol{x}_1, \boldsymbol{x}_2) = (\tilde{\boldsymbol{x}}_1, \tilde{\boldsymbol{x}}_2)} \in \mathbb{R}^{k_1}$$

For $f : \mathbb{R}^{k_1} \times \mathbb{R}^{k_2} \to \mathbb{R}^m, (\boldsymbol{x}_1, \boldsymbol{x}_2) \mapsto (f^1(\boldsymbol{x}_1, \boldsymbol{x}_2), \ldots, f^m(\boldsymbol{x}_1, \boldsymbol{x}_2))^\top \in \mathbb{R}^m$ we define (analogously w.r.t. second argument)

$$\text{Jacobian w.r.t. first argument:} \quad \frac{\partial f(\tilde{\boldsymbol{x}}_1, \tilde{\boldsymbol{x}}_2)}{\partial \boldsymbol{x}_1} = \left[ \frac{\partial f^i}{\partial x_1^j} \right]_{i,j} \Bigg|_{(\boldsymbol{x}_1, \boldsymbol{x}_2) = (\tilde{\boldsymbol{x}}_1, \tilde{\boldsymbol{x}}_2)} \in \mathbb{R}^{m \times k_1}$$

# B  PULLBACK GEOMETRY IN DIFFUSION MODELS

**Lemma B.1.** *Let $\mathcal{Z}$ be a latent space, $\mathcal{X} = \mathbb{R}^d$ a data space, and $f : \mathcal{Z} \to \mathcal{X}$ a decoder. Then the length and energy of a curve $\gamma : [0, 1] \to \mathcal{X}$ under the pullback geometry are given by*

$$\ell_{\mathrm{PB}}(\gamma) = \int_0^1 \left\| \tfrac{d}{ds} f(\gamma_s) \right\| ds \tag{24}$$

$$\mathcal{E}_{\mathrm{PB}}(\gamma) = \frac{1}{2} \int_0^1 \left\| \tfrac{d}{ds} f(\gamma_s) \right\|^2 ds, \tag{25}$$

*where $\| \cdot \|$ is the Euclidean norm.*

*Proof.* For a general Riemannian metric $\mathbf{G}$, the length, and energy are given by

$$\ell_{\mathbf{G}}(\gamma) = \int_0^1 \|\dot{\gamma}_s\|_{\mathbf{G}} ds \tag{26}$$

$$\mathcal{E}_{\mathbf{G}}(\gamma) = \frac{1}{2} \int_0^1 \|\dot{\gamma}_s\|_{\mathbf{G}}^2 ds, \tag{27}$$

where $\|\dot{\gamma}_s\|_{\mathbf{G}}^2 = \dot{\gamma}_s^\top G(\gamma_s)\dot{\gamma}_s$. For $\mathbf{G} = \mathbf{G}_{\mathrm{PB}}$ induced by the decoder $f$, we have

$$\mathbf{G}_{\mathrm{PB}}(z) = \left( \frac{\partial f}{\partial z}(z) \right)^\top \left( \frac{\partial f}{\partial z}(z) \right), \tag{28}$$

which leads to

$$\begin{aligned}
\|\dot{\gamma}_s\|_{\mathbf{G}_{\mathrm{PB}}}^2 &= \dot{\gamma}_s^\top \left( \frac{\partial f}{\partial z}(\gamma_s) \right)^\top \left( \frac{\partial f}{\partial z}(\gamma_s) \right) \dot{\gamma}_s = \left( \frac{\partial f}{\partial z}(\gamma_s)\dot{\gamma}_s \right)^\top \left( \frac{\partial f}{\partial z}(\gamma_s)\dot{\gamma}_s \right) \\
&\stackrel{(*)}{=} \left( \tfrac{d}{ds} f(\gamma_s) \right)^\top \left( \tfrac{d}{ds} f(\gamma_s) \right) = \left\| \tfrac{d}{ds} f(\gamma_s) \right\|^2,
\end{aligned} \tag{29}$$

where $(*)$ follows from the chain rule. $\qquad\square$

**Proposition B.1** (Pullback geodesics decode to straight lines). *Let $\mathcal{Z}$ be a latent space, $\mathcal{X} = \mathbb{R}^d$ a data space, and $f : \mathcal{Z} \to \mathcal{X}$ a bijective decoder. Fix $z^a, z^b \in \mathcal{Z}$ and write $x^a = f(z^a)$, $x^b = f(z^b)$. Then any shortest path between $z^a$ and $z^b$ in the pullback geometry decodes to the straight segment from $x^a$ to $x^b$.*

*Proof.* Let $x_s = (1 - s)\, x^a + s\, x^b$ for $s \in [0, 1]$. Because $f$ is bijective, define its latent preimage

$$\gamma_s^\star = f^{-1}(x_s), \qquad s \in [0, 1].$$

The pullback length of a latent curve $\gamma$ is the Euclidean length of its image (Lemma B.1):

$$\ell_{\mathrm{PB}}(\gamma) = \int_0^1 \left\| \tfrac{d}{ds} f(\gamma_s) \right\| ds.$$

For $\gamma^\star$, $f(\gamma_s^\star) = x_s$ has constant velocity $\dot{x}_s = x^b - x^a$, hence

$$\ell_{\mathrm{PB}}(\gamma^\star) = \int_0^1 \|x^b - x^a\| ds = \|x^b - x^a\|.$$

For any other smooth latent curve $\overline{\gamma}$ from $z^a$ to $z^b$, using the triangle inequality:

$$\ell_{\mathrm{PB}}(\overline{\gamma}) = \int_0^1 \left\| \tfrac{d}{ds} f(\overline{\gamma}_s) \right\| ds \geq \left\| \int_0^1 \tfrac{d}{ds} f(\overline{\gamma}_s)\, ds \right\| = \|f(z^b) - f(z^a)\| = \|x^b - x^a\| = \ell_f(\gamma^\star).$$

Therefore $\ell_{\mathrm{PB}}(\overline{\gamma}) \geq \ell_{\mathrm{PB}}(\gamma^\star)$. Hence, any pullback geodesic decodes to the straight segment:

$$f(\gamma_s^\star) = f(f^{-1}(x_s)) = (1 - s)\, x^a + s\, x^b.$$

$\qquad\square$

In this proposition, we emphasize that any minimizing path in the latent space $\mathcal{Z}$, when measured with the pullback metric, will always decode to a straight line in the data space $\mathcal{X}$. The reason is that the bijective decoder acts only on the ambient coordinates of $\mathcal{X} = \mathbb{R}^d$, regardless of whether the actual data lie on a lower-dimensional submanifold. In the denoising diffusion setting, this situation is unavoidable, since the model enforces $\dim(\mathcal{Z}) = \dim(\mathcal{X})$. This stands in contrast with prior works using variational autoencoders (Arvanitidis et al., 2018), where latent geodesics live in a lower-dimensional space and can decode to curved trajectories in the ambient space.

Unless the dimension of the latent space is reduced to the intrinsic dimension of the data, the pullback metric carries no meaningful geometric information in the standard denoising diffusion setting, where $\dim(\mathcal{Z}) = \dim(\mathcal{X})$.

## C  PROOF OF PROPOSITION 5.1

In this section, we prove Proposition 5.1. The proof consits of

1. Showing that curve energy $\mathcal{E}$ in exponential families simplifies (Appendix C.1).
2. Showing that the family of denoising distributions forms an exponential family (Appendix C.2).
3. Putting it together (Appendix C.4).

### C.1  INFORMATION GEOMETRY IN EXPONENTIAL FAMILIES

We begin by defining an exponential family of distributions.

**Definition C.1** (Exponential Family). *A parametric family of probability distributions $\{p(\cdot|\boldsymbol{z})\}$ is called an exponential family if it can be expressed in the form*

$$p(\boldsymbol{x}|\boldsymbol{z}) = h(\boldsymbol{x}) \exp\big(\boldsymbol{\eta}(\boldsymbol{z})^\top T(\boldsymbol{x}) - \psi(\boldsymbol{z})\big), \tag{30}$$

*with $\boldsymbol{x}$ a random variable modelling the data and $\boldsymbol{z}$ the parameter of the distribution. In addition, $T(\boldsymbol{x})$ is called a sufficient statistic, $\boldsymbol{\eta}(\boldsymbol{z})$ natural (canonical) parameter, $\psi(\boldsymbol{z})$ the log-partition (cumulant) function and $h(\boldsymbol{x})$ is a base measure independent of $\boldsymbol{z}$, and*

$$\boldsymbol{\mu}(\boldsymbol{z}) = \mathbb{E}\left[T(\boldsymbol{x}) \mid \boldsymbol{z}\right] \tag{31}$$

*is the expectation parameter.*

In exponential families, the Riemannian metric tensor takes a specific form, which we show now.

**Proposition C.1** (Fisher-Rao metric for an exponential family). *Let $\{p(\cdot|\boldsymbol{z})\}$ be an exponential family. We denote $\boldsymbol{\eta}(\boldsymbol{z})$ the natural parametrisation, $T(\boldsymbol{x})$ the sufficient statistic and $\boldsymbol{\mu}(\boldsymbol{z}) = \mathbb{E}[T(\boldsymbol{x})|\boldsymbol{z}]$ the expectation parameters. The Fisher-Rao metric is given by:*

$$\mathbf{G}_{\mathrm{IG}}(\boldsymbol{z}) = \left(\frac{\partial \boldsymbol{\eta}(\boldsymbol{z})}{\partial \boldsymbol{z}}\right)^\top \left(\frac{\partial \boldsymbol{\mu}(\boldsymbol{z})}{\partial \boldsymbol{z}}\right). \tag{32}$$

*Proof.* For $p(\boldsymbol{x}|\boldsymbol{z}) = h(\boldsymbol{x}) \exp\big(\boldsymbol{\eta}(\boldsymbol{z})^\top T(\boldsymbol{x}) - \psi(\boldsymbol{z})\big)$, we have

$$\nabla_{\boldsymbol{z}} \log p(\boldsymbol{x}|\boldsymbol{z}) = \nabla_{\boldsymbol{z}} \sum_k \eta^k(\boldsymbol{z}) T^k(\boldsymbol{x}) - \nabla_{\boldsymbol{z}} \psi(\boldsymbol{z}) = \left(\frac{\partial \boldsymbol{\eta}(\boldsymbol{z})}{\partial \boldsymbol{z}}\right)^\top T(\boldsymbol{x}) - \nabla_{\boldsymbol{z}} \psi(\boldsymbol{z}). \tag{33}$$

Note that

$$\mathbb{E}\left[\nabla_{\boldsymbol{z}} \log p(\boldsymbol{x}|\boldsymbol{z}) \mid \boldsymbol{z}\right] = \int p(\boldsymbol{x}|\boldsymbol{z}) \nabla_{\boldsymbol{z}} \log p(\boldsymbol{x}|\boldsymbol{z}) d\boldsymbol{x} = \int \nabla_{\boldsymbol{z}} p(\boldsymbol{x}|\boldsymbol{z}) d\boldsymbol{x} = \nabla_{\boldsymbol{z}} \int p(\boldsymbol{x}|\boldsymbol{z}) d\boldsymbol{x} = \mathbf{0}. \tag{34}$$

Therefore, by taking the expectation of both sides of Eq. 33, we get

$$\nabla_{\boldsymbol{z}} \psi(\boldsymbol{z}) = \left(\frac{\partial \boldsymbol{\eta}(\boldsymbol{z})}{\partial \boldsymbol{z}}\right)^\top \boldsymbol{\mu}(\boldsymbol{z}), \tag{35}$$

where $\boldsymbol{\mu}(\boldsymbol{z}) = \mathbb{E}[T(\boldsymbol{x})|\boldsymbol{z}]$. Now we differentiate $j$-th component of both sides of Eq. 34 w.r.t $z^i$, and we get

$$
\begin{aligned}
0 = \frac{\partial}{\partial z^i}0 &= \frac{\partial}{\partial z^i}\mathbb{E}\left[\frac{\partial \log p(\boldsymbol{x}|\boldsymbol{z})}{\partial z^j}\ \bigg|\ \boldsymbol{z}\right] = \frac{\partial}{\partial z^i}\int p(\boldsymbol{x}|\boldsymbol{z})\frac{\partial \log p(\boldsymbol{x}|\boldsymbol{z})}{\partial z^j}d\boldsymbol{x}\\
&= \int \frac{\partial p(\boldsymbol{x}|\boldsymbol{z})}{\partial z^i}\frac{\partial \log p(\boldsymbol{x}|\boldsymbol{z})}{\partial z^j}d\boldsymbol{x} + \int p(\boldsymbol{x}|\boldsymbol{z})\frac{\partial^2 \log p(\boldsymbol{x}|\boldsymbol{z})}{\partial z^i \partial z^j}d\boldsymbol{x}\\
&= \mathbb{E}\left[\frac{\partial \log p(\boldsymbol{x}|\boldsymbol{z})}{\partial z^i}\frac{\partial \log p(\boldsymbol{x}|\boldsymbol{z})}{\partial z^j}\ \bigg|\ \boldsymbol{z}\right] + \mathbb{E}\left[\frac{\partial^2 \log p(\boldsymbol{x}|\boldsymbol{z})}{\partial z^i \partial z^j}\ \bigg|\ \boldsymbol{z}\right].
\end{aligned}
\tag{36}
$$

Therefore

$$
[\mathbf{G}_{\text{IG}}(\boldsymbol{z})]_{ij} = \mathbb{E}\left[\frac{\partial \log p(\boldsymbol{x}|\boldsymbol{z})}{\partial z^i}\frac{\partial \log p(\boldsymbol{x}|\boldsymbol{z})}{\partial z^j}\ \bigg|\ \boldsymbol{z}\right] = -\mathbb{E}\left[\frac{\partial^2 \log p(\boldsymbol{x}|\boldsymbol{z})}{\partial z^i \partial z^j}\ \bigg|\ \boldsymbol{z}\right].
\tag{37}
$$

Now using Eq. 33, we have

$$
\frac{\partial^2 \log p(\boldsymbol{x}|\boldsymbol{z})}{\partial z^i \partial z^j} = \frac{\partial}{\partial z^i}\left(\sum_k \frac{\partial \eta^k(\boldsymbol{z})}{\partial z^j}T^k(\boldsymbol{x}) - \frac{\partial \psi(\boldsymbol{z})}{\partial z^j}\right) = \sum_k \frac{\partial^2 \eta^k(\boldsymbol{z})}{\partial z^i \partial z^j}T^k(\boldsymbol{x}) - \frac{\partial^2 \psi(\boldsymbol{z})}{\partial z^i \partial z^j}.
\tag{38}
$$

Therefore, from Eq. 37:

$$
[\mathbf{G}_{\text{IG}}(\boldsymbol{z})]_{ij} = \frac{\partial^2 \psi(\boldsymbol{z})}{\partial z^i \partial z^j} - \sum_k \frac{\partial^2 \eta^k(\boldsymbol{z})}{\partial z^i \partial z^j}\mu^k(\boldsymbol{z}).
\tag{39}
$$

Now using Eq. 35, we have

$$
\frac{\partial^2 \psi(\boldsymbol{z})}{\partial z^i \partial z^j} = \frac{\partial}{\partial z^j}\left(\sum_k \frac{\partial \eta^k(\boldsymbol{z})}{\partial z^i}\mu^k(\boldsymbol{z})\right) = \sum_k \frac{\partial^2 \eta^k(\boldsymbol{z})}{\partial z^j \partial z^i}\mu^k(\boldsymbol{z}) + \sum_k \frac{\partial \eta^k(\boldsymbol{z})}{\partial z^i}\frac{\partial \mu^k(\boldsymbol{z})}{\partial z^j}.
\tag{40}
$$

Combining (Eq. 39) with (Eq. 40) yields:

$$
[\mathbf{G}_{\text{IG}}(\boldsymbol{z})]_{ij} = \sum_k \frac{\partial \eta^k(\boldsymbol{z})}{\partial z^j}\frac{\partial \mu^k(\boldsymbol{z})}{\partial z^i} = \left[\left(\frac{\partial \boldsymbol{\eta}(\boldsymbol{z})}{\partial \boldsymbol{z}}\right)^\top\left(\frac{\partial \boldsymbol{\mu}(\boldsymbol{z})}{\partial \boldsymbol{z}}\right)\right]_{ij}.
\tag{41}
$$

$\square$

**Corollary C.1** (Energy function for an exponential family). *Let $\boldsymbol{\gamma} : [0,1] \to \mathcal{Z}$ be a smooth curve in the parameter space $\mathcal{Z}$ of an exponential family. Then*

$$
\mathcal{E}_{\text{IG}}(\boldsymbol{\gamma}) = \frac{1}{2}\int_0^1 \left(\tfrac{d}{ds}\boldsymbol{\eta}(\boldsymbol{\gamma}_s)\right)^\top\left(\tfrac{d}{ds}\boldsymbol{\mu}(\boldsymbol{\gamma}_s)\right)ds
\tag{42}
$$

$$
\ell_{\text{IG}}(\boldsymbol{\gamma}) = \int_0^1 \sqrt{\left(\tfrac{d}{ds}\boldsymbol{\eta}(\boldsymbol{\gamma}_s)\right)^\top\left(\tfrac{d}{ds}\boldsymbol{\mu}(\boldsymbol{\gamma}_s)\right)}ds.
\tag{43}
$$

*For a curve discretized uniformly into $N$ points, we have $s_n := \frac{n}{N-1}$, $\boldsymbol{\gamma}_n := \boldsymbol{\gamma}(s_n)$, and $\boldsymbol{\mu}_n := \boldsymbol{\mu}(\boldsymbol{\gamma}_n)$, $\boldsymbol{\eta}_n := \boldsymbol{\eta}(\boldsymbol{\gamma}_n)$, we have*

$$
\mathcal{E}_{\text{IG}}(\boldsymbol{\gamma}) \approx \frac{N-1}{2}\sum_{n=0}^{N-2}(\boldsymbol{\eta}_{n+1} - \boldsymbol{\eta}_n)^\top(\boldsymbol{\mu}_{n+1} - \boldsymbol{\mu}_n)
\tag{44}
$$

$$
\ell_{\text{IG}}(\boldsymbol{\gamma}) \approx \sum_{n=0}^{N-2}\sqrt{(\boldsymbol{\eta}_{n+1} - \boldsymbol{\eta}_n)^\top(\boldsymbol{\mu}_{n+1} - \boldsymbol{\mu}_n)}
\tag{45}
$$

*Proof.* The energy of $\boldsymbol{\gamma}$ is given by $\mathcal{E}_{\text{IG}}(\boldsymbol{\gamma}) = \frac{1}{2}\int_0^1 \|\dot{\boldsymbol{\gamma}}_s\|_{\mathbf{G}_{\text{IG}}}^2 ds$. We replace the Riemannian metric $\mathbf{G}_{\text{IG}}$ with the previously obtained expression of the Fisher-Rao metric (Eq. 32).

Using Eq. (32)), we have

$$\|\dot{\boldsymbol{\gamma}}_s\|^2_{\mathbf{G}_{\mathrm{IG}}} = \dot{\boldsymbol{\gamma}}_s^\top \mathbf{G}_{\mathrm{IG}}(\boldsymbol{\gamma}_s)\dot{\boldsymbol{\gamma}}_s = \dot{\boldsymbol{\gamma}}_s^\top \left(\frac{\partial \boldsymbol{\eta}(\boldsymbol{\gamma}_s)}{\partial \boldsymbol{z}}\right)^\top \left(\frac{\partial \boldsymbol{\mu}(\boldsymbol{\gamma}_s)}{\partial \boldsymbol{z}}\right)\dot{\boldsymbol{\gamma}}_s$$

$$= \left(\frac{\partial \boldsymbol{\eta}(\boldsymbol{\gamma}_s)}{\partial \boldsymbol{z}}\dot{\boldsymbol{\gamma}}_s\right)^\top \left(\frac{\partial \boldsymbol{\mu}(\boldsymbol{\gamma}_s)}{\partial \boldsymbol{z}}\dot{\boldsymbol{\gamma}}_s\right) = \left(\tfrac{d}{ds}\boldsymbol{\eta}(\boldsymbol{\gamma}_s)\right)^\top \left(\tfrac{d}{ds}\boldsymbol{\mu}(\boldsymbol{\gamma}_s)\right).$$

Therefore

$$\mathcal{E}_{\mathrm{IG}}(\boldsymbol{\gamma}) = \frac{1}{2}\int_0^1 \|\dot{\boldsymbol{\gamma}}_s\|^2_{\mathbf{G}_{\mathrm{IG}}} ds = \frac{1}{2}\int_0^1 \left(\tfrac{d}{ds}\boldsymbol{\eta}(\boldsymbol{\gamma}_s)\right)^\top \left(\tfrac{d}{ds}\boldsymbol{\mu}(\boldsymbol{\gamma}_s)\right) ds$$

$$\ell_{\mathrm{IG}}(\boldsymbol{\gamma}) = \int_0^1 \|\dot{\boldsymbol{\gamma}}_s\|_{\mathbf{G}_{\mathrm{IG}}} ds = \int_0^1 \sqrt{\left(\tfrac{d}{ds}\boldsymbol{\eta}(\boldsymbol{\gamma}_s)\right)^\top \left(\tfrac{d}{ds}\boldsymbol{\mu}(\boldsymbol{\gamma}_s)\right)} ds.$$

For a discretized curve, the reasoning is similar to the proof of Proposition A.2 in Arvanitidis et al. (2022). We have $ds = s_{n+1} - s_n = \frac{1}{N-1}$ and we can approximate $\frac{d}{ds}\boldsymbol{\eta}(\boldsymbol{\gamma}_s))\big|_{s=s_n} \approx \frac{\boldsymbol{\eta}_{n+1}-\boldsymbol{\eta}_n}{ds}$ and $\frac{d}{ds}\boldsymbol{\mu}(\boldsymbol{\gamma}_s))\big|_{s=s_n} \approx \frac{\boldsymbol{\mu}_{n+1}-\boldsymbol{\mu}_n}{ds}$, and

$$\mathcal{E}_{\mathrm{IG}}(\boldsymbol{\gamma}) \approx \frac{1}{2}\sum_{n=0}^{N-2}\left(\frac{\boldsymbol{\eta}_{n+1}-\boldsymbol{\eta}_n}{ds}\right)^\top \left(\frac{\boldsymbol{\mu}_{n+1}-\boldsymbol{\mu}_n}{ds}\right) ds = \frac{N-1}{2}\sum_{n=0}^{N-2}(\boldsymbol{\eta}_{n+1}-\boldsymbol{\eta}_n)^\top (\boldsymbol{\mu}_{n+1}-\boldsymbol{\mu}_n)$$

$$\ell_{\mathrm{IG}}(\boldsymbol{\gamma}) \approx \sum_{n=0}^{N-2}\sqrt{\left(\frac{\boldsymbol{\eta}_{n+1}-\boldsymbol{\eta}_n}{ds}\right)^\top \left(\frac{\boldsymbol{\mu}_{n+1}-\boldsymbol{\mu}_n}{ds}\right)} ds = \sum_{n=0}^{N-2}\sqrt{(\boldsymbol{\eta}_{n+1}-\boldsymbol{\eta}_n)^\top (\boldsymbol{\mu}_{n+1}-\boldsymbol{\mu}_n)}$$

$\square$

## C.2 DIFFUSION DENOISING DISTRIBUTIONS ARE EXPONENTIAL

A key observation is that the family of denoising distributions $p(\boldsymbol{x}_0|\boldsymbol{x}_t)$ indexed by both space and time $(\boldsymbol{x}_t, t)$ is exponential, which we prove now.

**Proposition C.2** (Exponential family of denoising). *Let $\boldsymbol{x}_t$ be a noisy observation corresponding to diffusion time $t$, as introduced in Eq. 1. Then the denoising distribution can be written as*

$$p(\boldsymbol{x}_0 \mid \boldsymbol{x}_t) = h(\boldsymbol{x}_0)\exp\left(\boldsymbol{\eta}(\boldsymbol{x}_t, t)^\top T(\boldsymbol{x}_0) - \psi(\boldsymbol{x}_t, t)\right), \tag{46}$$

*with $h = q$ the data distribution density, $\psi$ the log-partition function, and*

$$\boldsymbol{\eta}(\boldsymbol{x}_t, t) = \left(\frac{\alpha_t}{\sigma_t^2}\boldsymbol{x}_t, -\frac{\alpha_t^2}{2\sigma_t^2}\right) \quad \textit{(natural parameter)} \tag{47}$$

$$T(\boldsymbol{x}_0) = (\boldsymbol{x}_0, \|\boldsymbol{x}_0\|^2) \qquad \textit{(sufficient statistic)} \tag{48}$$

$$\boldsymbol{\mu}(\boldsymbol{x}_t, t) = \Big(\underbrace{\mathbb{E}\big[\boldsymbol{x}_0|\boldsymbol{x}_t\big]}_{\text{'space'}}, \underbrace{\frac{\sigma_t^2}{\alpha_t}\mathrm{div}_{\boldsymbol{x}_t}\mathbb{E}[\boldsymbol{x}_0|\boldsymbol{x}_t] + \big\|\mathbb{E}[\boldsymbol{x}_0|\boldsymbol{x}_t]\big\|^2}_{\text{'time'}: \ \mathbb{E}\big[\|\boldsymbol{x}_0\|^2 \mid \boldsymbol{x}_t\big]}\Big), \tag{49}$$

*which means that the family of denoising distributions $\{p(\boldsymbol{x}_0 \mid \boldsymbol{x}_t)\}$ indexed by $(\boldsymbol{x}_t, t)$ is exponential (Definition C.1).*

*Proof.* **Step 1: denoising is exponential.** The denoising distribution is given by

$$p(\boldsymbol{x}_0|\boldsymbol{x}_t) = \frac{p(\boldsymbol{x}_t|\boldsymbol{x}_0)q(\boldsymbol{x}_0)}{p_t(\boldsymbol{x}_t)},$$

where $q$ is the data distribution, $p(\boldsymbol{x}_t|\boldsymbol{x}_0) = \mathcal{N}(\boldsymbol{x}_t|\alpha_t\boldsymbol{x}_0, \sigma_t^2\boldsymbol{I})$ is the forward density (Eq. 1), and $p_t(\boldsymbol{x}_t) = \int p(\boldsymbol{x}_t|\boldsymbol{x}_0)q(\boldsymbol{x}_0)d\boldsymbol{x}_0$ is the marginal distribution at time $t$. Therefore

$$p(\boldsymbol{x}_t|\boldsymbol{x}_0) = \frac{1}{(2\pi\sigma_t^2)^{D/2}}\exp\left(-\frac{\|\boldsymbol{x}_t - \alpha_t\boldsymbol{x}_0\|^2}{2\sigma_t^2}\right)$$

$$= \frac{1}{(2\pi\sigma_t^2)^{D/2}}\exp\left(-\frac{\|\boldsymbol{x}_t\|^2}{2\sigma_t^2} + \frac{\alpha_t}{\sigma_t^2}\boldsymbol{x}_t^\top\boldsymbol{x}_0 - \frac{\alpha_t^2}{2\sigma_t^2}\|\boldsymbol{x}_0\|^2\right) \tag{50}$$

$$= \exp\left(-\frac{D}{2}\log(2\pi\sigma_t^2) - \frac{\|\boldsymbol{x}_t\|^2}{2\sigma_t^2}\right)\exp\left(-\frac{\alpha_t^2}{2\sigma_t^2}\|\boldsymbol{x}_0\|^2 + \frac{\alpha_t}{\sigma_t^2}\boldsymbol{x}_t^\top\boldsymbol{x}_0\right).$$

By substituting into the denoising density, we get

$$p(\boldsymbol{x}_0|\boldsymbol{x}_t) = q(\boldsymbol{x}_0)\exp\left\{-\frac{\alpha_t^2}{2\sigma_t^2}\|\boldsymbol{x}_0\|^2 + \frac{\alpha_t}{\sigma_t^2}\boldsymbol{x}_t^\top\boldsymbol{x}_0 - \left(\log p_t(\boldsymbol{x}_t) + \frac{D}{2}\log(2\pi\sigma_t^2) + \frac{\|\boldsymbol{x}_t\|^2}{2\sigma_t^2}\right)\right\}$$
$$= h(\boldsymbol{x}_0)\exp\left(\boldsymbol{\eta}(\boldsymbol{x}_t,t)^\top T(\boldsymbol{x}_0) - \psi(\boldsymbol{x}_t,t)\right),$$

(51)

where

$$\boldsymbol{\eta}(\boldsymbol{x}_t,t) = \left(\frac{\alpha_t}{\sigma_t^2}\boldsymbol{x}_t, -\frac{\alpha_t^2}{2\sigma_t^2}\right) \qquad \in \mathbb{R}^{D+1} \tag{52}$$

$$T(\boldsymbol{x}_0) = \left(\boldsymbol{x}_0, \|\boldsymbol{x}_0\|^2\right) \qquad \in \mathbb{R}^{D+1} \tag{53}$$

$$h(\boldsymbol{x}_0) = q(\boldsymbol{x}_0) \qquad \in \mathbb{R} \tag{54}$$

$$\psi(\boldsymbol{x}_t,t) = \log p_t(\boldsymbol{x}_t) + \frac{D}{2}\log(2\pi\sigma_t^2) + \frac{\|\boldsymbol{x}_t\|^2}{2\sigma_t^2} \qquad \in \mathbb{R} \tag{55}$$

which proves that the denoising distributions form an exponential family.

**Step 2: deriving the expectation parameter $\boldsymbol{\mu}$.** The expectation parameter $\boldsymbol{\mu}$ is given by $\boldsymbol{\mu}(\boldsymbol{x}_t,t) = \mathbb{E}\left[T(\boldsymbol{x}_0) \mid \boldsymbol{x}_t\right] = \left(\mathbb{E}[\boldsymbol{x}_0 \mid \boldsymbol{x}_t], \mathbb{E}[\|\boldsymbol{x}_0\|^2 \mid \boldsymbol{x}_t]\right)$. The denoising covariance is known (Meng et al., 2021):

$$\mathrm{Cov}[\boldsymbol{x}_0 \mid \boldsymbol{x}_t] = \frac{\sigma_t^2}{\alpha_t^2}\left(\boldsymbol{I} + \sigma_t^2\nabla_{\boldsymbol{x}_t}^2 \log p_t(\boldsymbol{x}_t)\right). \tag{56}$$

Therefore, from the definition of conditional variance, we can deduce the second denoising moment:

$$\mathbb{E}\left[\|\boldsymbol{x}_0\|^2 \mid \boldsymbol{x}_t\right] = \mathbb{E}\left[\left\|\boldsymbol{x}_0 - \mathbb{E}[\boldsymbol{x}_0 \mid \boldsymbol{x}_t]\right\|^2 \mid \boldsymbol{x}_t\right] + \left\|\mathbb{E}[\boldsymbol{x}_0 \mid \boldsymbol{x}_t]\right\|^2$$
$$= \mathrm{Tr}\left(\mathrm{Cov}[\boldsymbol{x}_0 \mid \boldsymbol{x}_t]\right) + \left\|\mathbb{E}[\boldsymbol{x}_0 \mid \boldsymbol{x}_t]\right\|^2$$
$$\stackrel{(56)}{=} \frac{\sigma_t^2}{\alpha_t^2}\left(D + \sigma_t^2\Delta\log p_t(\boldsymbol{x}_t)\right) + \left\|\mathbb{E}[\boldsymbol{x}_0 \mid \boldsymbol{x}_t]\right\|^2 \tag{57}$$
$$= \frac{\sigma_2^2}{\alpha_t}\mathrm{div}_{\boldsymbol{x}_t}\left(\frac{\boldsymbol{x}_t + \sigma_t^2\nabla_{\boldsymbol{x}_t}\log p_t(\boldsymbol{x}_t)}{\alpha_t}\right) + \left\|\mathbb{E}[\boldsymbol{x}_0 \mid \boldsymbol{x}_t]\right\|^2$$
$$= \frac{\sigma_t^2}{\alpha_t}\mathrm{div}_{\boldsymbol{x}_t}\mathbb{E}[\boldsymbol{x}_0 \mid \boldsymbol{x}_t] + \left\|\mathbb{E}[\boldsymbol{x}_0 \mid \boldsymbol{x}_t]\right\|^2,$$

where we used the fact that (Efron, 2011)

$$\mathbb{E}[\boldsymbol{x}_0 \mid \boldsymbol{x}_t] = \frac{\boldsymbol{x}_t + \sigma_t^2\nabla_{\boldsymbol{x}_t}\log p_t(\boldsymbol{x}_t)}{\alpha_t}. \tag{58}$$

Together, we have

$$\boldsymbol{\mu}(\boldsymbol{x}_t,t) = \left(\mathbb{E}\left[\boldsymbol{x}_0 \mid \boldsymbol{x}_t\right], \frac{\sigma_t^2}{\alpha_t}\mathrm{div}_{\boldsymbol{x}_t}\mathbb{E}[\boldsymbol{x}_0|\boldsymbol{x}_t] + \left\|\mathbb{E}[\boldsymbol{x}_0 \mid \boldsymbol{x}_t]\right\|^2\right) \tag{59}$$

$\square$

### C.3 BOLTZMANN DENOISING DISTRIBUTIONS

Note that, if the data distribution is Boltzmann, i.e. $q(\boldsymbol{x}_0) \propto \exp(-U(\boldsymbol{x}_0))$ for some energy function $U$, we have:

$$p(\boldsymbol{x}_0|\boldsymbol{x}_t) \propto q(\boldsymbol{x}_0)p(\boldsymbol{x}_t|\boldsymbol{x}_0) \propto \exp(-(U(\boldsymbol{x}_0))\exp\left(-\frac{\|\boldsymbol{x}_t - \alpha_t\boldsymbol{x}_0\|^2}{2\sigma_t^2}\right)$$
$$= \exp\left(-U(\boldsymbol{x}_0) - \tfrac{1}{2}\mathrm{SNR}(t)\|\boldsymbol{x}_0 - \boldsymbol{x}_t/\alpha_t\|^2\right).$$

This implies that $p(\boldsymbol{x}_0|\boldsymbol{x}_t)$ is also a Boltzmann distribution with $p(\boldsymbol{x}_0|\boldsymbol{x}_t) \propto \exp(-U(\boldsymbol{x}_0|\boldsymbol{x}_t))$ for

$$U(\boldsymbol{x}_0|\boldsymbol{x}_t) = U(\boldsymbol{x}_0) + \tfrac{1}{2}\mathrm{SNR}(t)\left\|\boldsymbol{x}_0 - \boldsymbol{x}_t/\alpha_t\right\|^2. \tag{60}$$

## C.4 PUTTING IT TOGETHER: PROPOSITION 5.1

The claim of Proposition 5.1 follows from Proposition C.2 and Corollary C.1.

## D KULLBACK-LEIBLER DIVERGENCE IN EXPONENTIAL FAMILIES

For any distribution family, the Fisher-Rao metric is the local approximation of the KL divergence, i.e (Arvanitidis et al., 2022):

$$\mathrm{KL}(p(\cdot|\boldsymbol{z}_1)||p(\cdot|\boldsymbol{z}_2)) \approx \frac{1}{2}(\boldsymbol{z}_1 - \boldsymbol{z}_2)^\top \mathbf{G}_{\mathrm{IG}}(\boldsymbol{z}_1)(\boldsymbol{z}_1 - \boldsymbol{z}_2).$$

In the case of exponential families, we have $\mathbf{G}_{\mathrm{IG}}(\boldsymbol{z}) = \left(\frac{\partial \boldsymbol{\eta}(\boldsymbol{z})}{\partial \boldsymbol{z}}\right)^\top \left(\frac{\partial \boldsymbol{\mu}(\boldsymbol{z})}{\partial \boldsymbol{z}}\right)$, and thus we can write

$$\mathrm{KL}(p(\cdot|\boldsymbol{z}_1)||p(\cdot|\boldsymbol{z}_2)) \approx \frac{1}{2}(\boldsymbol{z}_1 - \boldsymbol{z}_2)^\top \left(\frac{\partial \boldsymbol{\eta}(\boldsymbol{z}_1)}{\partial \boldsymbol{z}}\right)^\top \left(\frac{\partial \boldsymbol{\mu}(\boldsymbol{z}_1)}{\partial \boldsymbol{z}}\right)(\boldsymbol{z}_1 - \boldsymbol{z}_2)$$

$$\approx \frac{1}{2}(\boldsymbol{\eta}(\boldsymbol{z}_1) - \boldsymbol{\eta}(\boldsymbol{z}_2))^\top (\boldsymbol{\mu}(\boldsymbol{z}_1) - \boldsymbol{\mu}(\boldsymbol{z}_2)).$$

It turns out that the RHS always corresponds to a notion of distribution divergence (not only when $\boldsymbol{z}_1$ and $\boldsymbol{z}_2$ are close together), namely the *symmetrized* Kullback-Leibler divergence:

$$\mathrm{KL}^{\mathrm{S}}(p||q) := \frac{1}{2}(\mathrm{KL}(p||q) + \mathrm{KL}(q||p)). \tag{61}$$

**Lemma D.1** (KL in exponential families). *Let $\mathcal{P} = \{p(\cdot \mid \boldsymbol{z}) \mid \boldsymbol{z} \in \mathcal{Z}\}$ be an exponential family with $p(\boldsymbol{x}|\boldsymbol{z}) = h(\boldsymbol{x})\exp(\boldsymbol{\eta}(\boldsymbol{z})^\top T(\boldsymbol{x}) - \psi(\boldsymbol{z}))$, and $\boldsymbol{\mu}(\boldsymbol{z}) = \mathbb{E}_{\boldsymbol{x}\sim p(\boldsymbol{x}\mid\boldsymbol{z})}[T(\boldsymbol{x})]$. Then*

$$\mathrm{KL}(\boldsymbol{z}_1||\boldsymbol{z}_2) = (\boldsymbol{\eta}(\boldsymbol{z}_1) - \boldsymbol{\eta}(\boldsymbol{z}_2))^\top \boldsymbol{\mu}(\boldsymbol{z}_1) - \psi(\boldsymbol{z}_1) + \psi(\boldsymbol{z}_2), \tag{62}$$

*where we abuse notation and write $\mathrm{KL}(\boldsymbol{z}_1||\boldsymbol{z}_2)$ instead of $\mathrm{KL}(p(\cdot \mid \boldsymbol{z}_1)||p(\cdot \mid \boldsymbol{z}_2))$.*

*Proof.*

$$\mathrm{KL}(\boldsymbol{z}_1||\boldsymbol{z}_2) = \mathbb{E}_{\boldsymbol{x}\sim p(\boldsymbol{x}\mid\boldsymbol{z}_1)}[\log p(\boldsymbol{x}|\boldsymbol{z}_1) - \log p(\boldsymbol{x}|\boldsymbol{z}_2)]$$

$$= \mathbb{E}_{\boldsymbol{x}\sim p(\boldsymbol{x}\mid\boldsymbol{z}_1)}[\boldsymbol{\eta}(\boldsymbol{z}_1)^\top T(\boldsymbol{x}) - \boldsymbol{\eta}(\boldsymbol{z}_2)^\top T(\boldsymbol{x}) - \psi(\boldsymbol{z}_1) + \psi(\boldsymbol{z}_2)]$$

$$= (\boldsymbol{\eta}(\boldsymbol{z}_1) - \boldsymbol{\eta}(\boldsymbol{z}_2))^\top \mathbb{E}_{\boldsymbol{x}\sim p(\boldsymbol{x}\mid\boldsymbol{z}_1)}[T(\boldsymbol{x})] - \psi(\boldsymbol{z}_1) + \psi(\boldsymbol{z}_2)$$

$$= (\boldsymbol{\eta}(\boldsymbol{z}_1) - \boldsymbol{\eta}(\boldsymbol{z}_2))^\top \boldsymbol{\mu}(\boldsymbol{z}_1) - \psi(\boldsymbol{z}_1) + \psi(\boldsymbol{z}_2).$$

$\square$

**Lemma D.2** (Symmetrized KL in exponential families). *With assumptions of Lemma D.1, we have*

$$\mathrm{KL}^{\mathrm{S}}(\boldsymbol{z}_1||\boldsymbol{z}_2) = \frac{1}{2}(\boldsymbol{\eta}(\boldsymbol{z}_1) - \boldsymbol{\eta}(\boldsymbol{z}_2))^\top (\boldsymbol{\mu}(\boldsymbol{z}_1) - \boldsymbol{\mu}(\boldsymbol{z}_2)). \tag{63}$$

*Proof.*

$$2\,\mathrm{KL}^{\mathrm{S}}(\boldsymbol{z}_1 \mid \boldsymbol{z}_2) = \mathrm{KL}(\boldsymbol{z}_1||\boldsymbol{z}_2) + \mathrm{KL}(\boldsymbol{z}_2||\boldsymbol{z}_1)$$

$$= (\boldsymbol{\eta}(\boldsymbol{z}_1) - \boldsymbol{\eta}(\boldsymbol{z}_2))^\top \boldsymbol{\mu}(\boldsymbol{z}_1) - \cancel{\psi(\boldsymbol{z}_1)} + \cancel{\psi(\boldsymbol{z}_2)} + (\boldsymbol{\eta}(\boldsymbol{z}_2) - \boldsymbol{\eta}(\boldsymbol{z}_1))^\top \boldsymbol{\mu}(\boldsymbol{z}_2) - \cancel{\psi(\boldsymbol{z}_2)} + \cancel{\psi(\boldsymbol{z}_1)}$$

$$= (\boldsymbol{\eta}(\boldsymbol{z}_1) - \boldsymbol{\eta}(\boldsymbol{z}_2))^\top (\boldsymbol{\mu}(\boldsymbol{z}_1) - \boldsymbol{\mu}(\boldsymbol{z}_2)).$$

$\square$

The formula for KL in Lemma D.1 is not useful in practice, because it requires knowing $\psi(\boldsymbol{z})$, which can be unknown or expensive to evaluate. However, the gradients with respect to both arguments depend only on $\boldsymbol{\eta}$ and $\boldsymbol{\mu}$.

**Lemma D.3** (KL gradients). *With assumptions of Lemma D.1, we have for any $z_1, z_2$*

$$\nabla_{z_1} \mathrm{KL}(z_1||z_2) = \frac{\partial \mu(z_1)}{\partial z}^\top (\eta(z_1) - \eta(z_2))$$

$$\nabla_{z_2} \mathrm{KL}(z_1||z_2) = \frac{\partial \eta(z_2)}{\partial z}^\top (\mu(z_2) - \mu(z_1)) \tag{64}$$

*Proof.* The proof is a straightforward calculation using Lemma D.1 and Eq. 35. We have

$$\nabla_{z_1} \mathrm{KL}(z_1||z_2) = \nabla_{z_1} \left( (\eta(z_1) - \eta(z_2))^\top \mu(z_1) - \psi(z_1) + \psi(z_2) \right)$$

$$= \frac{\partial \eta(z_1)}{\partial z}^\top \mu(z_1) + \frac{\partial \mu(z_1)}{\partial z}^\top (\eta(z_1) - \eta(z_2)) - \nabla_z \psi(z_1)$$

$$\overset{(35)}{=} \cancel{\frac{\partial \eta(z_1)}{\partial z}^\top \mu(z_1)} + \frac{\partial \mu(z_1)}{\partial z}^\top (\eta(z_1) - \eta(z_2)) - \cancel{\frac{\partial \eta(z_1)}{\partial z}^\top \mu(z_1)}$$

$$= \frac{\partial \mu(z_1)}{\partial z}^\top (\eta(z_1) - \eta(z_2))$$

and

$$\nabla_{z_2} \mathrm{KL}(z_1||z_2) = \nabla_{z_2} \left( (\eta(z_1) - \eta(z_2))^\top \mu(z_1) - \psi(z_1) + \psi(z_2) \right)$$

$$\overset{(35)}{=} -\frac{\partial \eta(z_2)}{\partial z}^\top \mu(z_1) + \frac{\partial \eta(z_2)}{\partial z}^\top \mu(z_2) = \frac{\partial \eta(z_2)}{\partial z}^\top (\mu(z_2) - \mu(z_1))$$

$\square$

Knowing the gradients allows for estimating the KL divergence along a curve without knowing $\psi$.

**Proposition D.1** (KL along a curve). *Let $\gamma : [0,1] \to \mathcal{Z}$ be a smooth denoising curve, and $z^* \in \mathcal{Z}$. Then:*

$$\mathrm{KL}(\gamma_s||z^*) = \mathrm{KL}(\gamma_0||z^*) + \int_0^s \left( \frac{d}{du}\mu(\gamma_u) \right)^\top (\eta(\gamma_u) - \eta(z^*))\, du$$

$$\mathrm{KL}(z^*||\gamma_s) = \mathrm{KL}(z^*||\gamma_0) + \int_0^s \left( \frac{d}{du}\eta(\gamma_u) \right)^\top (\mu(\gamma_u) - \mu(z^*))\, du \tag{65}$$

*Proof.*

$$\mathrm{KL}(\gamma_s||z^*) - \mathrm{KL}(\gamma_0||z^*) =$$

$$= \int_0^s \frac{d}{du} \left( \mathrm{KL}(\gamma_u||z^*) \right) du \qquad \text{// Fundamental theorem of calculus}$$

$$= \int_0^s \nabla_{z_1} \mathrm{KL}(\gamma_u||z^*)^\top \dot{\gamma}_u du \qquad \text{// Chain rule}$$

$$= \int_0^s \left( \frac{\partial \mu(\gamma_u)}{\partial z} \dot{\gamma}_u \right)^\top (\eta(\gamma_u) - \eta(z^*))\, du \qquad \text{// Lemma D.3}$$

$$= \int_0^s \left( \frac{d}{du}\mu(\gamma_u) \right)^\top (\eta(\gamma_u) - \eta(z^*))\, du \qquad \text{// Chain rule.}$$

Using the same reasoning we have

$$\mathrm{KL}(\boldsymbol{z}^*||\boldsymbol{\gamma}_s) - \mathrm{KL}(\boldsymbol{z}^*||\boldsymbol{\gamma}_0) = \int_0^s \frac{d}{du}\left(\mathrm{KL}(\boldsymbol{z}^*||\boldsymbol{\gamma}_u)\right) du$$

$$= \int_0^s \nabla_{\boldsymbol{z}_2}\mathrm{KL}(\boldsymbol{z}^*||\boldsymbol{\gamma}_u)^\top \dot{\boldsymbol{\gamma}}_u du$$

$$= \int_0^s \left(\frac{\partial \boldsymbol{\eta}(\boldsymbol{\gamma}_u)}{\partial \boldsymbol{z}}\dot{\boldsymbol{\gamma}}_u\right)^\top (\boldsymbol{\mu}(\boldsymbol{\gamma}_u) - \boldsymbol{\mu}(\boldsymbol{z}^*)) du$$

$$= \int_0^s \left(\tfrac{d}{du}\boldsymbol{\eta}(\boldsymbol{\gamma}_u)\right)^\top (\boldsymbol{\mu}(\boldsymbol{\gamma}_u) - \boldsymbol{\mu}(\boldsymbol{z}^*)) du$$

$\square$

## E  GEODESIC AND DIFFED PSEUDOCODE

---

**Algorithm 2** Evaluate Curve

---

**Require:** Curve $\boldsymbol{\gamma} : [0, 1] \to \mathbb{R}^{D+1}$, number of points $N_\gamma$
1: $\{s_n\}_{n=0}^{N_\gamma - 1} \leftarrow \text{linspace}(0, 1, N_\gamma)$          ▷ Uniformly discretized curve
2: **for** $n = 0$ to $N_\gamma - 1$ **do**
3:      $\boldsymbol{z}_n \leftarrow \boldsymbol{\gamma}(s_n)$
4:      Decompose $\boldsymbol{z}_n = (\boldsymbol{x}_t^{(n)}, t_n)$
5:      $\boldsymbol{\eta}_n \leftarrow \boldsymbol{\eta}(\boldsymbol{x}_t^{(n)}, t_n)$ using Eq. (15)
6:      $\boldsymbol{\mu}_n \leftarrow \boldsymbol{\mu}(\boldsymbol{x}_t^{(n)}, t_n)$ using Eq. (16)
7: **end for**
8: **return** $\{\boldsymbol{\eta}_n\}, \{\boldsymbol{\mu}_n\}$

---

**Algorithm 3** Spacetime Geodesic Estimation

---

**Require:** $\boldsymbol{x}_a, \boldsymbol{x}_b \in \mathbb{R}^D$, $N_\gamma > 0$ discretization points
**Require:** $N_{\text{iter}} > 0$, $t_{\min} > 0$, learning rate $\eta > 0$
1: $\boldsymbol{z}_a \leftarrow (\boldsymbol{x}_a, t_{\min})$, $\boldsymbol{z}_b \leftarrow (\boldsymbol{x}_b, t_{\min})$          ▷ Embed points into Spacetime
2: Initialize cubic spline $\boldsymbol{\gamma_\theta}$ with $\boldsymbol{\gamma_\theta}(0) = \boldsymbol{z}_a$, $\boldsymbol{\gamma_\theta}(1) = \boldsymbol{z}_b$
3: **for** $k = 0$ to $N_{\text{iter}} - 1$ **do**          ▷ Optimization loop
4:      $\{\boldsymbol{\eta}_n\}, \{\boldsymbol{\mu}_n\} \leftarrow \text{EVALUATECURVE}(\boldsymbol{\gamma_\theta}, N_\gamma)$
5:      $\mathcal{E}(\boldsymbol{\gamma_\theta}) \leftarrow \frac{N_\gamma - 1}{2} \sum_n (\boldsymbol{\eta}_{n+1} - \boldsymbol{\eta}_n)^\top (\boldsymbol{\mu}_{n+1} - \boldsymbol{\mu}_n)$      ▷ Energy estimate Eq. (14)
6:      $\boldsymbol{g} \leftarrow \nabla_{\boldsymbol{\theta}} \mathcal{E}(\boldsymbol{\gamma_\theta})$
7:      $\boldsymbol{\theta} \leftarrow \boldsymbol{\theta} - \eta \boldsymbol{g}$          ▷ Gradient descent update of curve's parameters
8: **end for**
9: **return** $\boldsymbol{\gamma_\theta}$

---

**Algorithm 4** Diffusion Edit Distance Estimation

---

**Require:** $\boldsymbol{x}_a, \boldsymbol{x}_b \in \mathbb{R}^D$, $N_\gamma > 0$, $N_{\text{iter}} > 0$
**Require:** $t_{\min} > 0$, learning rate $\eta > 0$
1: $\boldsymbol{\gamma_\theta} \leftarrow \text{SPACETIMEGEODESIC}(\boldsymbol{x}_a, \boldsymbol{x}_b, N_\gamma, N_{\text{iter}}, t_{\min}, \eta)$      ▷ Use Algorithm 3
2: $\{\boldsymbol{\eta}_n\}, \{\boldsymbol{\mu}_n\} \leftarrow \text{EVALUATECURVE}(\boldsymbol{\gamma_\theta}, N_\gamma)$
3: $d_{\text{DE}}(\boldsymbol{x}_a, \boldsymbol{x}_b) \leftarrow \sum_n \sqrt{(\boldsymbol{\eta}_{n+1} - \boldsymbol{\eta}_n)^\top (\boldsymbol{\mu}_{n+1} - \boldsymbol{\mu}_n)}$      ▷ Length estimate Eq. (45)
4: **return** $d_{\text{DE}}(\boldsymbol{x}_a, \boldsymbol{x}_b)$

---

## F  QUALITATIVE EXAMPLE OF DIFFED

In Fig. 8, we compare the Diffusion Edit Distance (DiffED) (Section 6.2) with LPIPS (Zhang et al., 2018), SSIM (Wang et al., 2004), and the Euclidean distance between images. Specifically, for the ImageNet class "Space bar", we generate 20 random image pairs, estimate the similarity of each pair with each method, and rank the pairs according to each method, from the most similar to the most dissimilar.

## G  EXPERIMENTAL DETAILS

### G.1  TOY GAUSSIAN MIXTURE

For the experiments with a 1D Gaussian mixture (Fig. 1, and Fig. 3 left), we define the data distribution as $p_0 = \sum_{i=1}^3 \pi_i \mathcal{N}(\mu_i, \sigma^2)$ with $\mu_1 = -2.5, \mu_2 = 0.5, \mu_3 = 2.5, \pi_1 = 0.275, \pi_2 = 0.45, \pi_3 = 0.275$, and $\sigma = 0.75$. We specify the forward process (Eq. 1) as Variance-Preserving (Song et al., 2021), i.e. satisfying $\alpha_t^2 + \sigma_t^2 = 1$, and assume as log-SNR linear noise schedule,

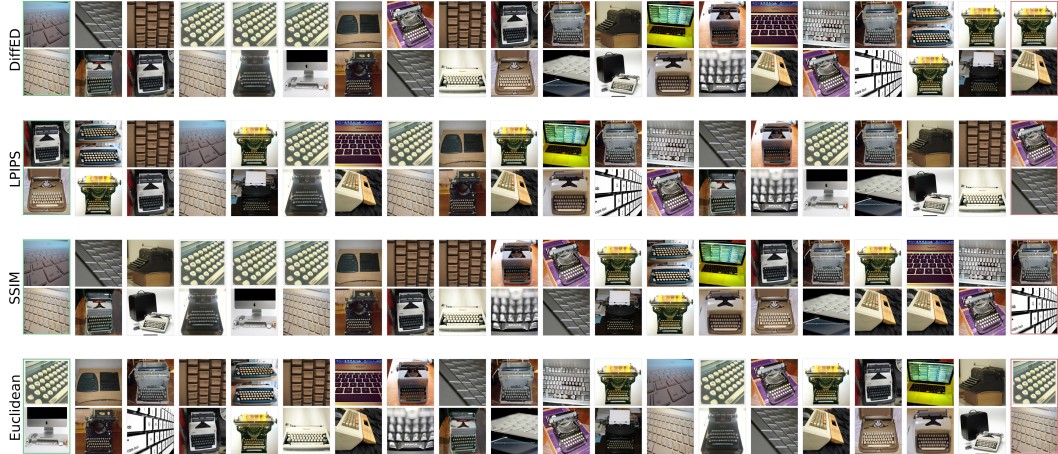

Figure 8: **Comparison of DiffED with other image similarity metrics**. Each row corresponds to a different image similarity measure, and images and sorted by their similarity, from most similar (left) to most dissimilar (right). Images shown are 20 random image pairs from class "Space bar".

i.e. $\lambda_t = \log \mathrm{SNR}(t) = \lambda_{\max} + (\lambda_{\min} - \lambda_{\max})t$ for $\lambda_{\min} = -10, \lambda_{\max} = 10$. Which implies: $\alpha_t^2 = \mathrm{sigmoid}(\lambda_t), \sigma_t^2 = \mathrm{sigmoid}(-\lambda_t)$.

Since $p_0$ is a Gaussian mixture, all marginals $p_t$ are also Gaussian mixtures, and training a diffusion model is unnecessary, as the score function $\nabla_{\boldsymbol{x}} \log_t(\boldsymbol{x})$ is known analytically. In this example, the data is 1D, and the spacetime is 2D.

To generate Fig. 1 we estimate the geodesic between $\boldsymbol{z}_1 = (-2.3, 0.35)$, and $\boldsymbol{z}_2 = (2, 0.4)$ by parametrizing $\boldsymbol{\gamma}$ with a cubic spline (Arvanitidis et al., 2022) with two nodes, and discretizing it into $N = 128$ points and taking 1000 optimization steps with Adam optimizer and learning rate $\eta = 0.1$, which takes a few seconds on an M1 CPU.

To generate Fig. 3 left, we generate 3 PF-ODE sampling trajectories starting from $x = 1, 0, -1$ using an Euler solver with 512 solver steps. We solve only until $t = t_{\min} = 0.1$ (as opposed to $t = 0$), because for $t \approx 0$, the denoising distributions $p(\boldsymbol{x}_0|\boldsymbol{x}_t)$ become closer to Dirac delta distributions $\delta_{\boldsymbol{x}_t}$, which makes the energies very large. For each sampling trajectory, we take the endpoints $(x_1, 1), (x_{t_{\min}}, t_{\min})$ and estimate the geodesic between them using Proposition 5.1 with a cubic spline with 10 nodes, discretizing it into 512 points, and taking 2000 gradient steps of AdamW optimizer with learning rate $\eta = 0.01$. This takes roughly 10 seconds on an M1 CPU.

### G.2 IMAGE DATA

For all experiments on image data, we use the pretrained EDM2 model trained on ImageNet512 (Karras et al., 2024) (specifically, the `edm2-img512-xxl-fid` checkpoint), which is a Variance-Exploding model, i.e. $\alpha_t = 1$, and using the noise schedule $\sigma_t = t$. It is a latent diffusion model, using a fixed StabilityVAE (Rombach et al., 2022) as the encoder/decoder.

**Image interpolations.** To interpolate between to images, we encode them with StabilityVAE to obtain two latent codes $\boldsymbol{x}_0^1, \boldsymbol{x}_0^2$, and encode them both with PF-ODE (Eq. 3) from $t = 0$ to $t = t_{\min} = 0.368$, corresponding to $\log \mathrm{SNR}(t_{\min}) = 2$. This is to avoid very high values of energy for $t \approx 0$. We then optimize the geodesic between $(\boldsymbol{x}_{t_{\min}}^1, t_{\min})$ and $(\boldsymbol{x}_{t_{\min}}^2, t_{\min})$ by parametrizing it with a cubic spline with 8 nodes, and minimizing the energy defined in Proposition 5.1 using AdamW optimizer with learning rate $\eta = 0.1$. The curve is discretized into 16 points, and optimized for 200 gradient steps, which takes roughly 6 minutes on an A100 NVIDIA GPU per interpolation image pair.

Note that in our experiments, we used the largest release model `edm2-img512-xxl-fid`. The image interpolation time can be reduced to roughly a minute by considering the smallest model version `edm2-img512-xs-fid`.

**PF-ODE sampling trajectories.** To generate PF-ODE sampling trajectories, we use the 2nd order Heun solver (Karras et al., 2022) with 64 steps, and solve from $t = 80$ to $t_{\min} = 0.135$ corresponding to $\log \text{SNR}(t_{\min}) = 4$. This is to avoid instabilities for small $t$. We parametrize the geodesic directly with the entire sampling trajectory $\boldsymbol{\gamma}_t = (\boldsymbol{x}_t, t)$ for $t = T, \ldots, t_{\min}$, where the $t$ schedule corresponds to EDM2 model's sampling schedule.

We then fix the endpoints of the trajectory, and optimize the intermediate points using AdamW optimizer with learning rate $\eta = 0.0001$ (larger learning rates lead to `NaN` values) and take 600 optimization steps. This procedure took roughly 2 hours on an A100 NVIDIA GPU per a single sampling trajectory.

To visualize intermediate noisy images at diffusion time $t$, we rescale them with $\frac{\sigma_{\text{data}}}{\sqrt{\sigma_{\text{data}}^2 + \sigma_t^2}}$ before decoding with the VAE deocoder, to avoid unrealistic color values, where we set $\sigma_{\text{data}} = 0.5$ as in Karras et al. (2022).

### G.3 MOLECULAR DATA

**Approximating the base energy function with a neural network.** We follow Holdijk et al. (2023) and represent the energy function of Alanine Dipeptide in the space of two dihedral angles $\phi, \psi \in [-\pi, \pi)$. We use the code provided by the authors at `github.com/LarsHoldijk/SOCTransitionPaths`, which estimates the energy $U(\phi, \psi)$. However, even though the values of the energy $U$ looked reasonably, we found that the provided implementation of $\frac{\partial U}{\partial \phi}$, and $\frac{\partial U}{\partial \psi}$ yielded unstable results due to discontinuities.

Instead, we trained an auxiliary feedforward neural network $U_\theta$ to approximate $U$. We parametrized with two hidden layers of size 64 with SiLU activation functions, and trained it on a uniformly discretized grid $[-\pi, \pi] \times [-\pi, \pi]$ into 16384 points. We trained the model with mean squared error for 8192 steps using Adam optimizer with a learning rate $\eta = 0.001$ until the model converged to an average loss of $\approx 1.5$. This took approximately two and a half minutes on an M1 CPU. In the subsequent experiments, we estimate $\nabla_{\boldsymbol{x}} U(\boldsymbol{x})$ with automatic differentiation on the trained auxiliary model.

**Generating samples from the energy landscape.** To generate samples from the data distribution $p_0(\boldsymbol{x}_0) \propto \exp(-U(\boldsymbol{x}_0))$, we initialize the samples uniformly on the $[-\pi, \pi] \times [-\pi, \pi]$ grid, and use Langevin dynamics

$$d\boldsymbol{x} = -\nabla_{\boldsymbol{x}} U(\boldsymbol{x})dt + \sqrt{2}dW_t \tag{66}$$

with the Euler-Maruyama solver for $dt = 0.001$ and $N = 1000$ steps.

**Training a diffusion model on the energy landscape.** To estimate the spacetime geodesics, we need a denoiser network approximating the denoising mean $\hat{\boldsymbol{x}}_0(\boldsymbol{x}_t, t) \approx \mathbb{E}[\boldsymbol{x}_0 | \boldsymbol{x}_t]$. We parametrize the denoiser network with

```
from ddpm import MLP
model = MLP(
    hidden_size=128,
    hidden_layers=3,
    emb_size=128,
    time_emb="sinusoidal",
    input_emb="sinusoidal"
)
```

using the TinyDiffusion implementation `github.com/tanelp/tiny-diffusion`. We trained the model using the weighted denoising loss: $w(\lambda_t)\|\hat{\boldsymbol{x}}_0(\boldsymbol{x}_t, t) - \boldsymbol{x}_0\|^2$ with a weight function $w(\lambda_t) = \sqrt{\text{sigmoid}(\lambda_t + 2)}$ and an adaptive noise schedule (Kingma & Gao, 2023). We train the model for 4000 steps using the AdamW optimizer with learning rate $\eta = 0.001$, which took roughly 1 minute on an M1 CPU.

**Spacetime geodesics.** With a trained denoiser $\hat{\boldsymbol{x}}_0(\boldsymbol{x}_t, t)$, we can estimate the expectation parameter $\boldsymbol{\mu}$ (Eq. 16) and thus curves energies in the spacetime geometry (Proposition 5.1).

In Section 6.3, we want to interpolate between two low-energy states: $\boldsymbol{x}_0^1 = (-2.55, 2.7)$ and $\boldsymbol{x}_0^2 = (0.95, -0.4)$. To avoid instabilities for $t \approx 0$, we represent them on the spacetime manifold as $\boldsymbol{z}_1 = (-2.55, 2.7, t_{\min})$, and $\boldsymbol{z}_2 = (0.95, -0.4, t_{\min})$, where $\log \mathrm{SNR}(t_{\min}) = 4$. We then approximate the geodesic between them by parametrizing $\boldsymbol{\gamma}$ as a cubic spline with 10 nodes and fixed endpoints $\boldsymbol{\gamma}_0 = \boldsymbol{z}_1$, and $\boldsymbol{\gamma}_1 = \boldsymbol{z}_2$ and discretize it into 128 points. We then optimize it by minimizing Proposition 5.1 with the Adam optimizer with learning rate $\eta = 0.1$ and take 10000 optimization steps, which takes roughly 4 minutes on an M1 CPU.

**Annealed Langevin dynamics.** To generate transition paths, we use Annealed Langevin dynamics (Algorithm 1) with the geodesic discretized into $N = 128$ points, $K = 128$ Langevin steps for each point on the geodesic $\boldsymbol{\gamma}$, and use $dt = 0.0004$, i.e., requiring 16385 evaluations of the gradient of the auxiliary energy function per path. Generating 1000 independent paths in parallel takes 32 seconds on an M1 CPU and requires a total of 16,385,000 energy function evaluations.

**Constrained transition paths.** Constrained transition paths were also parametrized with cubic splines with 10 nodes, but discretized into 1024 points.

For the **low-variance** transition paths, we chose the threshold $\rho = 3$, and $\lambda = 0$ for the first 1200 optimization steps, and $\lambda$ linearly increasing from 0 to 100 for the last 3800 optimization steps, for the total of 5000 optimization steps with the Adam optimizer with a learning $\eta = 0.01$. This took just under 6 minutes on an M1 CPU.

For the **region-avoiding** transition paths, we encode the restricted region with $\boldsymbol{z}^* = (-0.8, -0.1, t^*)$ with $\log \mathrm{SNR}(t^*) = 4$, and combine two penalty functions: $h_1$ is the low-variance penalty described above, but with $\rho_1 = 3.75$ threshold, and $h_2$ is the KL penalty with $\rho_2 = -4350$ threshold. We define $\lambda_1$ as in the low-variance transitions, and fix $\lambda_2 = 1$. The optimization was performed with Adam optimizer, learning rate $\eta = 0.1$, and ran for 4000 steps for a runtime of just under 5 minutes on an M1 CPU.

The reason we include the low-variance penalty in the region-avoiding experiment is because $\mathrm{KL}(p(\cdot|\boldsymbol{z}^*) \,||\, p(\cdot|\boldsymbol{\gamma}_s))$ can trivially be increased by simply increasing entropy of $p(\cdot|\boldsymbol{\gamma}_s)$ which would not result in avoiding the region defined by $p(\cdot|\boldsymbol{z}^*)$.

## H  NOTE ON TRANSITION PATH SAMPLING BASELINES

For transition path experiments performed in Section 6.3, we considered Holdijk et al. (2023); Du et al. (2024); Raja et al. (2025) as baselines. However, we encountered reproducibility issues. Specifically

- Holdijk et al. (2023) released the implementation: `github.com/LarsHoldijk/SOCTransitionPaths`. However, it does not appear to be supported. Several issues in the repository highlight failures to reproduce results, which have remained unresolved for more than a year.
- Raja et al. (2025) released the implementation: `github.com/ASK-Berkeley/OM-TPS`. However, it does not contain the code for the alanine dipeptide experiments, and the authors did not respond to a request to release it.
- Du et al. (2024) released the implementation: `github.com/plainerman/Variational-Doob` that we were able to use. However, we obtained results significantly worse than those reported in the original publication. We have contacted the authors, who acknowledged our question but did not provide guidance on how to resolve the issue.

For Doob's Lagrangian (Du et al., 2024), we experimented with: different numbers of epochs, different numbers of Gaussians, first vs second order ODE, MLP vs spline, and internal vs external coordinates. We reported the results of the configuration that was the best. Many configurations either diverged completely (returned `NaN` values) or collapsed to completely straight transition paths, oblivious to the underlying energy landscape. These issues persisted even after switching to double precision (as advised in the official code repository).

## I  EXPECTATION PARAMETER ESTIMATION CODE

```python
import jax
import jax.random as jr
import jax.numpy as jnp

def f(x, t, key): # Implemenation of the expected denoising
    pass

def sigma_and_alpha(t): # Depends on the choice of SDE and noise
    schedule
    pass

def mu(x, t, key):
    model_key, eps_key = jr.split(key, 2)
    eps = jr.rademacher(eps_key, (x.size,), dtype=jnp.float32)
    def pred_fn(x_):
        return f(x_, t, key=model_key)
    f_pred, f_grad = jax.jvp(pred_f, (x,), (eps,))
    div = jnp.sum(f_grad * eps)
    sigma, alpha = sigma_and_alpha(t)
    return sigma**2/alpha * div + jnp.sum(f_pred ** 2), f_pred
```

Listing 1: JAX Implementation of $\mu$ estimation

## J    LICENCES

- EDM2 model (Karras et al., 2024): Creative Commons BY-NC-SA 4.0 license
- ImageNet dataset (Deng et al., 2009): Custom non-commercial license
- SDVAE model (Rombach et al., 2022): CreativeML Open RAIL++-M license
- OpenM++ (OpenMP Architecture Review Board, 2008): MIT License

