# OpenReview forum: "The Spacetime of Diffusion Models: An Information Geometry Perspective"
_ICLR.cc/2026/Conference — ICLR 2026 Oral_

### Official Review · Reviewer_XMm1 · 2025-10-31

**Soundness:** 3
**Presentation:** 3
**Contribution:** 3
**Rating:** 6
**Confidence:** 4

**Summary:**

This paper presents an information-geometric perspective on the latent space of diffusion models. The authors first show that the conventional pullback metric induced by the Probability Flow ODE (PF-ODE) is degenerate: geodesics under this metric always decode to straight lines in data space, thus failing to capture any intrinsic geometry. To address this limitation, the authors propose a new Fisher–Rao metric defined on the denoising distributions $p(x_0 | x_t)$. They further introduce a latent spacetime representation $z = (x_t, t)$, where both the sample and its diffusion timestep jointly parameterize the geometry. They prove that these denoising distributions form an exponential family, which enables a tractable estimator for geodesic energy and distance. Based on this framework, they define the Diffusion Edit Distance (DiffED) and demonstrate applications in image interpolation and transition path sampling for molecular systems, showing competitive performance with specialized baselines.

**Strengths:**

- The paper provides a conceptually novel and rigorous framework.
- This paper shows the inherent limitations of the pullback metric defined by the PF-ODE sampler from $x\_{T}$ to $x\_{0}$ and motivates the introduction of the information geometry using the denoising distribution $p(x\_0 | x\_{T})$.
- The transition-path sampling experiments on the Alanine Dipeptide system demonstrate that the framework has potential beyond visualization or interpolation tasks.

**Weaknesses:**

- The claim that “memorylessness” causes $p(x\_0 | x\_T) \approx q(x\_0​)$ deserves clearer probabilistic justification. While the forward process is memoryless, it seems like this does not directly imply conditional independence in the reverse direction. Isn't it $p(x_0 | x_{T}) \propto p(x_{T}) q(x_0)$?
- Table 1 reports the number of energy evaluations but omits the actual trajectory computation time or runtime comparison with baselines.

**Questions:**

- Could the proposed Diffusion Edit Distance be leveraged for practical downstream applications, such as by combining with SDEdit?
- How would the transition path differ if one were to fix a specific diffusion timestep $t$ (as in Line 169) instead of employing the current spacetime modeling? What are the conceptual or empirical benefits of jointly modeling both space and time in the proposed spacetime representation? The current discussion in Lines 175–181 mainly explains how the framework enables spacetime geodesic modeling, but it does not clearly state why this joint modeling is beneficial or what additional insights it provides.

---

> ### Author Response · Authors · 2025-11-20
> **Response to Reviewer XMm1**
>
> We want to thank the Reviewer for their insightful and constructive review. Below, we address the raised concerns and questions.
>
> *The claim that “memorylessness” causes $p(x_0|x_T) \approx q(x_0)$ deserves clearer probabilistic justification. While the forward process is memoryless, it seems like this does not directly imply conditional independence in the reverse direction. Isn't it $p(x_0|x_T) \propto p(x_T)q(x_0)$?*
>
> We are happy to elaborate on this. From Bayes' rule, we have:
>
> $$p(x_0 | x_T)=\frac{p(x_T|x_0)q(x_0)}{p(x_T)} = q(x_0) \underbrace{\frac{p(x_T | x_0)}{p(x_T)}}_{\approx 1} \approx q(x_0).$$
>
> Intuitively, even though the "memorylessness" condition appears asymmetric, it implies that the two variables $x_0, x_T$ are independent. This means that their joint density factorizes $p(x_0, x_T) = q(x_0)p(x_T)$ and the conditionals in both directions $p(x_0|x_T)$ and $p(x_T|x_0)$ don't depend on what we condition on.
>
> *Table 1 reports the number of energy evaluations but omits the actual trajectory computation time or runtime comparison with baselines.*
>
> This is a valid point. The reason we follow related work and report the number of function evaluations is to provide more of an "apples to apples" comparison. For example, in our experiments, we used PyTorch on a CPU, while [1] uses a Jax-compiled model running on a GPU.
>
> Below, we provide the estimated wall-clock times of all methods on our Apple M1 CPU:
> * MCMC-fixed-length: ~72 hours
> * Doob's Lagrangian: ~45 hours (on an NVIDIA A100 40GB GPU: ~1 hour)
> * MCMC-variable-length ~70 minutes
> * Ours: ~10 minutes
>
> *Could the proposed Diffusion Edit Distance be leveraged for practical downstream applications, such as by combining with SDEdit?*
>
> We agree that this is an interesting direction, although we have not investigated it. We hypothesize that our geometric framework could be leveraged to improve methods like SDEdit. Instead of relying on a heuristically chosen noise level $t$ to corrupt the image, DiffED could potentially be used to infer the optimal, data-specific noise/corruption level required to minimize the transition cost between the source image and the target concept.
>
> *How would the transition path differ if one were to fix a specific diffusion timestep instead of employing the current spacetime modeling? What are the conceptual or empirical benefits of jointly modeling both space and time in the proposed spacetime representation? The current discussion mainly explains how the framework enables spacetime geodesic modeling, but it does not clearly state why this joint modeling is beneficial or what additional insights it provides.*
>
> These are good questions, and we appreciate the opportunity to elaborate. Modeling space and time jointly is inherently more general than fixing a single diffusion timestep. By constraining geodesics to move only within a fixed time slice, we force them onto a sub-optimal path that incurs a strictly higher energy cost than the true spacetime geodesics. As visualized in Figure 1, the true geodesic naturally varies significantly along the time axis to minimize this energy. Restricting the geometry to a fixed t removes this degree of freedom, resulting in sub-optimal trajectories where the corresponding denoising distributions change more rapidly (i.e., have higher energy).
>
> A major practical benefit of this joint formalism is the ability to estimate geodesics between clean data points. We identify a clean point $x_0$ with a denoising distribution at a very small $t_{min}$, which allows us to anchor geodesics at clean endpoints while still exploiting the flexibility of higher noise levels for the path in between. This mechanism is exactly what enables the Diffusion Edit Distance and the transition path sampling results in Section 6.3. Conversely, if we were to fix the diffusion time to a very small $t_{min}$ (to represent clean data), the geometry would collapse to the Euclidean metric (similar to the Pullback geometry case) and the geodesics would reduce to straight lines in data space.
>
> We are happy to add this additional discussion (and formally prove that the Fisher-Rao geometry for fixed $t$ collapses to Euclidean as $t\to 0$) in the final revision.
>
> ---
>
> [1] Du et al. "Doob’s Lagrangian: A Sample-Efficient Variational Approach to Transition Path Sampling" (NeurIPS 2024)
>
> ---
>
> We would like to thank the Reviewer again. We hope that our response clarified any doubts and convinced you to reconsider your score.

---

> ### Author Response · Authors · 2025-11-27
> **Gentle reminder**
>
> Dear Reviewer XMm1
>
> As the discussion deadline approaches, we wanted to check that our previous reply has addressed your concerns. Should any further questions or specific points remain, we will be happy to provide further clarification.
>
> We sincerely appreciate the time and detailed effort you have invested in reviewing our submission.

---

### Official Review · Reviewer_r6Rp · 2025-11-01

**Soundness:** 3
**Presentation:** 2
**Contribution:** 2
**Rating:** 4
**Confidence:** 3

**Summary:**

The paper proposes a new geometric framework for diffusion models based on information geometry, introducing a spacetime manifold that combines both the noisy sample and the diffusion time (x_t, t). The authors show that the conditional denoising distributions, $p(x_0|x_t)$, form an exponential family, allowing them to endow the diffusion process with a Fisher–Rao metric (a principled way to measure distances between nearby denoising distributions)

Using this metric, they define spacetime geodesics (shortest paths) between data points and introduce the Diffusion Edit Distance (DiffED), a new notion of distance induced by the geometry of the diffusion process. They compute these geodesics by discretizing the trajectory in spacetime and minimizing a Fisher–Rao energy functional using gradients derived from the trained denoiser.

Empirically, the paper illustrates the framework on image diffusion models and molecular simulation tasks, showing that the resulting geodesics correspond to meaningful transition paths and that the proposed distance can characterize structure in data manifolds.

**Strengths:**

- The paper addresses an interesting theoretical question: how to define a meaningful geometric structure for diffusion models and connects it to information geometry.

- The idea of modeling denoising distributions as an exponential family and equipping the resulting spacetime with a Fisher–Rao metric is conceptually sound and mathematically motivated.

- The second application (transition path sampling in molecular systems) is original and can lead to generating more research in that direction (with the caveat that I'm not very familiar with this particular area of applications of diffusion models)

**Weaknesses:**

- The paper does not clearly describe the algorithm used to compute the Diffusion Edit Distance (DiffED). Including pseudocode or a concise algorithmic summary would greatly improve clarity and reproducibility.

- The computational cost of finding geodesics in spacetime using DiffED is unclear. Since Equation (16) must be evaluated at many noise levels, the method appears potentially expensive. How does its efficiency compare to related approaches? For instance, What’s Inside Your Diffusion Model? For example in [1]  the authors propose a similar framework for computing geodesics with diffusion models. Some discussion or comparison would help. This paper also contains some comparisons with other methods which the authors can use.

*A Score-Based Riemannian Metric to Explore the Data Manifold, Azeglio & Di Bernardo, arXiv:2505.11128

- The paper notes that the proposed distance correlates with SSIM but not with LPIPS. Could the authors elaborate on why this occurs, given that LPIPS is generally regarded as a stronger measure of perceptual or semantic similarity than SSIM?

Minor comments:

- Some terminology seems unnecessarily reinvented. For example, using “denoising” or “decoding distribution” instead of the standard term posterior may reduce clarity for readers familiar with Bayesian or diffusion-model terminology.

- Correlations are reported as percentages in line 281. should these not be expressed as standard correlation coefficients?

- The caption of Figure 1 describes the geodesic in spacetime as the “shortest path between two distributions.” Shouldn’t it instead be the shortest path between two images (or corresponding distributions over clean images)?

**Questions:**

- What is the motivation for comparing to the ODE baseline? It is not clear what specific insight this comparison is intended to convey.

- The pullback geodesic experiment seems somewhat like a strawman comparison—does it add significant insight? It may not warrant an entire page of the paper.

- Is the time discretization used to compute the geodesic linear, or is it adapted in some way along the path?

- How do the proposed results compare to stochastic interpolants? A direct comparison or discussion would be helpful for positioning this work relative to recent diffusion-based interpolation methods.

- What theoretical or empirical guarantees can you provide that the Diffusion Edit Distance (DiffED) indeed corresponds to a true geodesic under your defined Fisher–Rao metric?

---

> ### Author Response · Authors · 2025-11-20
> **Response to Reviewer r6Rp (1/2)**
>
> We thank the Reviewer for their thorough review and helpful questions. Below, we address the raised concerns.
>
> > *The paper does not clearly describe the algorithm used to compute the Diffusion Edit Distance (DiffED). Including pseudocode or a concise algorithmic summary would greatly improve clarity and reproducibility.*
>
> We agree that presenting the pseudocode directly in the paper is beneficial. We made the following updates to the paper
>
> * Include the pseudocode for both geodesic estimation and diffusion edit distance computation in Appendix E. We also added references to them in the main text.
> * Extended Corollary C.1 to show how we approximate both the energy and length of discretized curves.
>
> and the full implementation remains in the supplementary material for reference.
>
> > *The computational cost of finding geodesics in spacetime using DiffED is unclear. Since Equation (16) must be evaluated at many noise levels, the method appears potentially expensive. How does its efficiency compare to related approaches? For example, in [1] the authors propose a similar framework for computing geodesics with diffusion models. Some discussion or comparison would help. This paper also contains some comparisons with other methods which the authors can use.*
>
> We thank the Reviewer for bringing this reference to our attention. Both our method and [1] scale similarly, as they require discretizing a curve into N points and updating parameters via gradient descent. Our energy estimation requires N Jacobian-Vector Products (JVPs) of the score model per step, while the method in [1] requires N standard score network calls. Although a JVP is computationally roughly twice as expensive as a standard forward pass, our method converges in fewer iterations, requiring approximately 200 gradient steps compared to the 2000 steps reported in [1]. Consequently, we estimate our method to be roughly 5× faster in wall-clock time, though we note that a precise benchmark would require identical model architectures and hardware. We acknowledge that both our approach and [1] are computationally more intensive than simple heuristics like SLERP.
>
> Beyond computational cost, we would also like to clarify a fundamental distinction between the frameworks. While [1] defines a geometry on the noisy latent space for a fixed noise level $t$, our approach constructs a geometry over the entire spacetime continuum $z=(x_t​,t)$. This generalization is critical: it allows us to define a proper distance metric between clean data points (Diffusion Edit Distance), a feature not available when geometry is restricted to a single noise level. Furthermore, the spacetime framework generalizes beyond image interpolation to tasks such as transition path sampling in molecular systems.
>
> We are happy to include this comparison and discussion in the final version of the manuscript.
>
> > *The paper notes that the proposed distance correlates with SSIM but not with LPIPS. Could the authors elaborate on why this occurs, given that LPIPS is generally regarded as a stronger measure of perceptual or semantic similarity than SSIM?*
>
> We believe this is in line with the intuition that our metric roughly measures “how much noise needs to be added to image1 before it can be denoised to image2”. Adding small amounts of noise mainly affects fine-grained details (high frequencies), whilst keeping the high-level features (low frequencies) unchanged. Therefore, we believe that our metric might be more suitable for applications where SSIM is favoured over LPIPS, such as super-resolution or image deblurring.
>
> > *Some terminology seems unnecessarily reinvented. For example, using “denoising” or “decoding distribution” instead of the standard term posterior may reduce clarity for readers familiar with Bayesian or diffusion-model terminology.*
>
> Good point. Even though we find use of "denoising distribution" e.g., in [2], this terminology is indeed less common than we originally thought. However, we find that the term "posterior" can also be ambiguous in the diffusion literature. For example, in [3] the "approximate posterior" refers to $q(x_1, \dots, x_T | x_0)$  and "forward process posterior" means $q(x_{t-1}|x_t, x_0)$.
>
> We will clarify in the paper by noting that $p(x_0|x_t)$ is the posterior distribution of the data given a noisy $x_t$, or simply the "denoising distribution".
>
> > *Correlations are reported as percentages in line 281. should these not be expressed as standard correlation coefficients?*
>
> We are happy to remove the percentage notation if this improves clarity.

---

> ### Author Response · Authors · 2025-11-20
> **Response to Reviewer r6Rp (2/2)**
>
> > *The caption of Figure 1 describes the geodesic in spacetime as the “shortest path between two distributions.” Shouldn’t it instead be the shortest path between two images (or corresponding distributions over clean images)?*
>
> You are correct that these paths connect corresponding distributions over clean data. Since each spacetime point $z=(x,t)$ defines a posterior $p(x_0​∣x_t​)$, we refer to the geodesic as a path between these distributions to emphasize the probabilistic nature of the interpolation. For each point on the curve $x_t$, we visualize the corresponding posterior $p(x_0|x_t)$ on the left.
>
> > *What is the motivation for comparing to the ODE baseline? It is not clear what specific insight this comparison is intended to convey. The pullback geodesic experiment seems somewhat like a strawman comparison—does it add significant insight? It may not warrant an entire page of the paper.*
>
> The comparison to the ODE baseline serves to bridge our work with the extensive existing literature on the geometry of generative models. Most prior work defines Riemannian metrics via pullbacks through **deterministic** decoders $f:Z→X$ (e.g., [4]). The Probability Flow ODE provides the natural analog for diffusion models to attempt this standard approach.
>
> The insight conveyed by this comparison is a negative but crucial one: it demonstrates that the standard deterministic pullback approach is ill-suited for diffusion models due to the injectivity of the ODE map (as discussed in Section 4). This failure explicitly motivates our shift to the **stochastic** decoder view, showing that a proper geometric treatment of diffusion models requires accounting for their stochastic nature, rather than treating them as deterministic maps.
>
> > *Is the time discretization used to compute the geodesic linear, or is it adapted in some way along the path?*
>
> Yes, we uniformly discretize the curve. We have now made it explicit in the pseudocode in Algorithm 2 in the revised manuscript.
>
> > *How do the proposed results compare to stochastic interpolants? A direct comparison or discussion would be helpful for positioning this work relative to recent diffusion-based interpolation methods.*
>
> The reason we did not include a direct comparison is that our primary objective differs from typical interpolation methods. We were not aiming to generate photorealistic intermediate images, but rather to explore the intrinsic geometric structure of the diffusion latent space. Consequently, as shown in Figure 4, our geodesics naturally traverse through "noisy" regions of the space, meaning the intermediate steps are not intended to look like realistic, clean images by design.
>
> Instead, the value of our approach lies in defining meaningful distances - something that is missing in standard baselines. For instance, methods like SLERP lack a metric structure entirely, while geometric approaches like [1] operate at a fixed noise level, implicitly defining distances between noisy points rather than clean data. Finally, this geometric structure opens up applications beyond image interpolation, such as molecular transition path sampling, which standard diffusion-based interpolation methods do not offer.
>
> > *What theoretical or empirical guarantees can you provide that the Diffusion Edit Distance (DiffED) indeed corresponds to a true geodesic under your defined Fisher–Rao metric?*
>
> Since DiffED is simply the length of the approximate geodesic (Equation 17, and Algorithm 4 in the updated manuscript), we can guarantee that $\gamma$ is indeed a geodesic, but we cannot guarantee that this will always correspond to the length of a globally shortest path. Please see our general response for more details.
>
> ---
>
> [1] Azeglio et al. "WHAT’S INSIDE YOUR DIFFUSION MODEL? A SCORE-BASED RIEMANNIAN METRIC TO EXPLORE THE DATA MANIFOLD (ArXiv 2025)
>
> [2] Song et al. "Maximum Likelihood Training of Score-Based Diffusion Models" (NeurIPS 2021)
>
> [3] Ho et al. "Denoising Diffusion Probabilistic Models" (NeurIPS 2020)
>
> [4] Arvanitidis et al. "Latent Space Oddity: on the Curvature of Deep Generative Models" (ICLR 2018)
>
> ---
>
> We thank the Reviewer again for their time and constructive feedback. We hope that our response has sufficiently addressed your concerns, and we would be grateful if you would reconsider your assessment.

---

> ### Author Response · Authors · 2025-11-27
> **Gentle reminder**
>
> Dear Reviewer r6Rp,
>
> As the deadline for the discussion approaches, we wanted to follow up to ensure that our responses have adequately addressed your concerns. If there are any additional questions or points for discussion, we would be happy to address them promptly.
>
> We look forward to your feedback and sincerely appreciate your time and effort in reviewing our work.

---

### Official Review · Reviewer_ixrc · 2025-11-01

**Soundness:** 3
**Presentation:** 4
**Contribution:** 4
**Rating:** 8
**Confidence:** 5

**Summary:**

The authors extend a geometric perspective on diffusion models by defining the Fisher–Rao metric not only over the latent space but across the latent spacetime $(z = (x_t, t))$. This structure allows them to define the length, or energy, of curves within the latent spacetime, and to define geodesics, or shortest path, over latents and timesteps. The authors show that geodesics from noise $(t = T)$ to denoised states $(t = 0)$ closely follow PF-ODE sampling trajectories. This also leads to a new measure of distance, the Diffusion Edit Distance, expanding upon known interpolation distances such as LPIPS.

**Strengths:**

The paper is clearly written, and the proposed approach is well presented.

The work introduces a novel and interesting notion of latent spacetime. While previous studies usually considered interpolation between samples as transitions within a slice of latent space fixed in timestep, this paper generalizes that perspective. It may have a broad impact on the image editing domain, effectively connecting geometric ideas with techniques such as DDIM inversion.

**Weaknesses:**

There are concerns regarding the computational complexity. Although the paper mentions that the method may be slower, it would be beneficial to include exact runtime comparisons and additional evaluations.


The authors demonstrate that PF-ODE sampling trajectories and geodesics are similar but do not provide any quantitative metrics. It would be interesting to see how this approach compares with LPIPS, for example by fixing the starting and final points to the same timestep.

**Questions:**

The paper could benefit from discussion on image editing approaches such as DDIM and Null-text inversion.

The authors mention that their method is numerically slower and suggest potential distillation. What are their preliminary thoughts on the architecture and training objective for a separate model intended to predict DiffED? Specifically, should the distillation target focus on replicating the absolute geodesic length, or rather on the manifold’s tangent structure defined by the Fisher–Rao metric, to ensure geometrically meaningful similarity estimation?

---

> ### Author Response · Authors · 2025-11-20
> **Response to Reviewer ixrc**
>
> We thank the reviewer for their support of our work and helpful feedback. Below, we address the raised concerns.
>
> > *There are concerns regarding the computational complexity. Although the paper mentions that the method may be slower, it would be beneficial to include exact runtime comparisons and additional evaluations.*
>
> This is a fair point. We address this question in more detail in our response to Reviewer zLiS. In a nutshell: 1) Our approach is more efficient than available baselines in the molecular transition path sampling experiments; 2) It is faster than some geometric-based approaches to interpolations (such as [1]); 3) but it is orders of magnitude slower than traditional image metrics like SSIM or LPIPS.
>
> We will include these comparisons and make the efficiency statements more precise in the updated version of the manuscript.
>
> > *The authors demonstrate that PF-ODE sampling trajectories and geodesics are similar but do not provide any quantitative metrics. It would be interesting to see how this approach compares with LPIPS, for example, by fixing the starting and final points to the same timestep.*
>
> We thank the Reviewer for this suggestion. We agree that our claim in Section 6.1 regarding the visual similarity of PF-ODE trajectories and spacetime geodesics would benefit from quantitative verification.
>
> Following your recommendation, we performed an additional experiment where we fixed the start point ($x_T​$) and end point ($x_0$) to be identical for both the PF-ODE sampling trajectory and the spacetime geodesic. We then measured LPIPS between the corresponding states of the two trajectories across all intermediate timesteps.
>
> We found that the LPIPS distance remains extremely low throughout the entire process, starting at 0 (by design) and reaching a maximum of approximately 0.045. This confirms our qualitative claim that the two paths are perceptually nearly indistinguishable.
>
> We will include these additional results in the revised manuscript.
>
> > *The paper could benefit from discussion on image editing approaches such as DDIM and Null-text inversion.*
>
> This is a good point. We discuss the potential benefits of using our geometric framework together with SDEdit in our response to Reviewer XMm1, and we are happy to extend this discussion with DDIM and null-text inversion in the final version of the paper. While this investigation was out of scope for this project, we agree that this is an interesting direction for future work.
>
> > *The authors mention that their method is numerically slower and suggest potential distillation. What are their preliminary thoughts on the architecture and training objective for a separate model intended to predict DiffED? Specifically, should the distillation target focus on replicating the absolute geodesic length, or rather on the manifold’s tangent structure defined by the Fisher–Rao metric, to ensure geometrically meaningful similarity estimation?*
>
> We appreciate the suggestion regarding the manifold’s tangent structure. Our initial plan was a lightweight convolutional architecture - avoiding the massive data and compute requirements of contrastive objectives like CLIP - trained via direct regression to predict precomputed DiffED scalar values. However, we agree that targeting the tangent structure defined by the Fisher–Rao metric could yield a richer similarity estimation. For example, analyzing the eigendecomposition of the metric tensor could identify 'frequency-like' semantic directions that maximize/minimize the change in the denoising distribution. A key technical challenge to addressing this in a distillation framework is that the Fisher-Rao metric is most informative at non-zero noise levels, becoming unstable near clean data as the posterior $p(x_0​∣x_t​)$ approaches a Dirac delta. Consequently, while a scalar regression on total geodesic length is the most straightforward starting point, a geometrically faithful auxiliary model would likely require supervision to match the metric's properties at intermediate spacetime representations ($t>0$) rather than solely at the endpoints.
>
> ---
>
> [1] Azeglio et al. "WHAT’S INSIDE YOUR DIFFUSION MODEL? A SCORE-BASED RIEMANNIAN METRIC TO EXPLORE THE DATA MANIFOLD (ArXiv 2025)
>
> ---
>
> We would like to thank the Reviewer again for their time and effort put into this review, and we hope that our response answered all questions.

---

### Official Review · Reviewer_zLiS · 2025-11-03

**Soundness:** 4
**Presentation:** 3
**Contribution:** 3
**Rating:** 8
**Confidence:** 4

**Summary:**

This paper revisits the geometric structure underlying diffusion models and argues that meaningful geometry does not reside in the noisy latent space x_t alone, but rather in the full spacetime domain formed by pairs (x_t, t) combining state and noise level. The authors show that when considering only the latent variable x_t at a fixed noise level, the denoising distribution p(x_0 \mid x_t) becomes increasingly isotropic as t increases, causing the geometry (e.g., tangent and normal decomposition, pullback metrics) to collapse. To recover a non-trivial Riemannian structure, the paper proposes defining the metric using the Fisher–Rao information geometry over spacetime, resulting in the GIG metric that varies jointly with state and time.

Using this metric, the authors define the Diffusion Edit Distance (DiffED); the length of the shortest geodesic path in spacetime connecting two clean samples (x,0) and (y,0). This yields a principled notion of semantic distance that measures “how naturally one data point can be transformed into another” under the diffusion generative process. Moreover, the paper introduces a transition-path sampling method based on Annealed Langevin Dynamics (ALD) that follows the computed spacetime geodesic to generate actual interpolation trajectories. Empirical results in Section 6 demonstrate that these spacetime geodesics produce smooth, consistent, and semantically meaningful transformations between examples.

**Strengths:**

1. Clear and compelling reframing of the latent representation in diffusion models.
While prior work typically analyzes individual x_t states or only the fully-noised x_T, this paper emphasizes that the entire trajectory \{x_t\} constitutes the latent representation. Defining geometry on the spacetime manifold (x_t, t) using the Fisher–Rao metric is both conceptually straightforward and surprisingly underexplored, making the contribution feel natural yet novel.
2. Diffusion Edit Distance offers meaningful semantic comparisons.
The proposed DiffED provides a principled notion of distance based on how naturally one data point can transform into another through the diffusion process. Although currently computationally expensive, the definition itself seems robust and opens opportunities for future research in efficient approximations and downstream applications such as morphing, retrieval, and generative editing.
3. Transition-path sampling via Annealed Langevin Dynamics adds practical value.
The use of ALD to follow the spacetime geodesic demonstrates that the metric is not only theoretically motivated but also operationally useful. The method enables the generation of smooth and interpretable interpolation paths, giving the framework concrete applicability rather than remaining purely abstract.

**Weaknesses:**

1. High computational cost.
The method requires computing geodesics in spacetime and then sampling along those paths with ALD. This is quite expensive in practice, and the paper does not propose any way to reduce this cost. As a result, it may be difficult to use the method in large-scale or time-sensitive settings.
2. Limited variety of data types in experiments.
The experiments are mostly on datasets where the structure is relatively simple and smooth (e.g., faces, digits). It is unclear how well this approach works on more complex images (e.g., multiple objects, cluttered scenes) or on stylized data (e.g., cartoons, anime). Testing or discussing these cases would make the evaluation stronger.
3. Unclear stability of geodesic optimization.
The final interpolation path comes from optimizing an energy function, which may have multiple local minima. The paper does not analyze how sensitive the result is to initialization or parameter choices. Because of this, it is not yet clear how stable or reproducible the method is.

**Questions:**

1. Comparing image pairs with different visual attributes:
For two cases (i) two images with similar color tone but completely different content, and (ii) two images with identical structure/shape but very different color, which pair does the Diffusion Edit Distance assign a larger distance to? In other words, does the metric primarily reflect semantic structure, appearance, or a combination of both?

1-1. Potential analytical uses of DiffED.
Since Diffusion Edit Distance provides a meaningful notion of distance in the spacetime geometry, it seems possible to use it for analyzing model behavior (e.g., structure of learned manifolds, semantic neighborhood relationships, mode connectivity). Have the authors explored such analytical applications, or do they have ideas for experiments where DiffED could be used as a diagnostic tool?

2. Extension to video data:
Do the authors believe that the proposed spacetime geodesic framework can be extended to video sequences? If so, what would be a reasonable way to handle the temporal consistency constraint, for example, by treating time as an additional dimension alongside the diffusion timestep, or by defining geodesics directly in a trajectory space?

3. Effect of different latent representations.
The behavior of DiffED may depend on the type of latent space in which the diffusion model operates (e.g., VAE latent space, pixel space, or non-image modalities). Have the authors observed different geometric patterns or performance characteristics when applying the method to models operating in these different latent representations?

---

> ### Author Response · Authors · 2025-11-20
> **Response to Reviewer zLiS (1/2)**
>
> We thank the Reviewer for their support of our work and interesting suggestions. Below, we address the raised concerns.
>
> > *High computational cost. The method requires computing geodesics in spacetime and then sampling along those paths with ALD. This is quite expensive in practice, and the paper does not propose any way to reduce this cost. As a result, it may be difficult to use the method in large-scale or time-sensitive settings.*
>
> We take this opportunity to clarify that the Annealed Langevin Dynamics (ALD) sampler is only used in the molecular transition path sampling experiments (when we want to generate a diverse set of continuous transition paths), and as we have now provided a detailed runtime comparison in our response to Reviewer XMm1, **our method is more efficient than available baselines**. See below the wall clock times on an M1 Apple CPU
> * MCMC-fixed-length: ~72 hours
> * Doob's Lagrangian: ~45 hours (on an NVIDIA A100 40GB GPU: ~1 hour)
> * MCMC-variable-length ~70 minutes
> * Ours: ~10 minutes
>
> For estimating the Diffusion Edit Distance, we do not use ALD, but we still need to estimate the spacetime geodesic. As we discuss in the Appendix with experiment details, the cost of finding the geodesic for a single image pair ranges from under one minute for the smallest model to around 6 minutes for the largest one. This is faster than 20 minutes reported by another geometric-based approach [1], but still orders of magnitude slower than simple image metrics, whose runtime (on an h200 GPU) is approximately
> * LPIPS (VGG): 1.9ms
> * LPIPS (AlexNet): 1.2ms
> * SSIM (torchmetrics): 0.4ms
>
> Therefore, as we discuss in our limitations section, the proposed diffusion distance metric is not yet suitable for applications where it needs to be evaluated very often (e.g., as a loss function).
>
> > *Limited variety of data types in experiments. The experiments are mostly on datasets where the structure is relatively simple and smooth (e.g., faces, digits). It is unclear how well this approach works on more complex images (e.g., multiple objects, cluttered scenes) or on stylized data (e.g., cartoons, anime).*
>
> It is true that we did not explore stylized data as an application, but the ImageNet dataset that we used in our experiment does contain diverse structures (beyond faces or digits).
>
> > *Unclear stability of geodesic optimization. The final interpolation path comes from optimizing an energy function, which may have multiple local minima. The paper does not analyze how sensitive the result is to initialization or parameter choices. Because of this, it is not yet clear how stable or reproducible the method is.*
>
> This is a good point. We address this in detail in our general response. In a nutshell, our procedure for finding a geodesic (described in Algorithm 3 of the updated manuscript) is guaranteed to find a geodesic as long as it converges. In general, we cannot guarantee that this is a global minimum of the energy functional, but it is always a locally shortest path. Please see the general response for more details.
>
> > *Comparing image pairs with different visual attributes: [...] does the metric primarily reflect semantic structure, appearance, or a combination of both?*
>
> It is a very good question. We are still working on this, and we hope to have more results by the end of the discussion period.
>
> > *it seems possible to use it for analyzing model behavior. Have the authors explored such analytical applications, or do they have ideas for experiments where DiffED could be used as a diagnostic tool*
>
> This is a very good question. In theory, DiffED depends only on the specification of the (forward) diffusion process and the data distribution. This is unlike LPIPS, which depends on the learned features, so strongly depends on the model architecture (as evidenced by the fact that LPIPS(AlexNet) does not correlate very strongly with LPIPS(VGG)).
>
> Therefore, we believe that this metric has the potential to be used to analyze the semantic topology of the data distribution without the bias of auxiliary classifiers. Furthermore, outliers or out-of-distribution samples would likely exhibit disproportionately high Diffusion Edit Distances to their nearest neighbors in the training set, as the model would require a more 'expensive' sequence of edits to generate them.

---

> ### Author Response · Authors · 2025-11-20
> **Reponse to Reviewer zLiS (2/2)**
>
> > *Extension to video data: [...] what would be a reasonable way to handle the temporal consistency constraint*
>
> This is an interesting question. The proposed framework is defined independently of the modality (as long as it's continuous), and as such can in theory be applied to video sequences. The formulation would be oblivious to the video-temporal axis and simply treat data as higher-dimensional. Therefore, a geodesic would connect video sequences, and every intermediate point on a geodesic would correspond to a distribution over video sequences. The video-temporal constraint should be handled "automatically" by the fact that each $p(x_0|x_t)$ (where $x_0$ is a clean video and $x_t$ is a noisy video) should have this property if the base model is accurate enough.
>
> An interesting question specific to video modelling would be the extension of the proposed geometric framework to handle Diffusion Forcing [2], where the model allows different levels of noise for different frames. We have not investigated this direction.
>
> > *Effect of different latent representations [...] latent space, pixel space, non-image modalities [...]*
>
> Regarding latent vs pixel space, we only explored latent diffusion, as this is the prevailing paradigm (EDM, SD, FLUX). As in [3], we hypothesize that it should not make a qualitative difference, as the "latent representation" can largely be interpreted as a lower-resolution version of the image. In terms of different modalities, we expect that audio would be an interesting use case, especially since "adding noise" means different things in the wave domain and in the frequency domain. Therefore, the resulting distance would depend on the chosen representation. This analysis was also out of scope for our project.
>
> ---
>
> [1] Azeglio et al. "WHAT’S INSIDE YOUR DIFFUSION MODEL? A SCORE-BASED RIEMANNIAN METRIC TO EXPLORE THE DATA MANIFOLD (ArXiv 2025)
>
> [2] Chen et al. "Diffusion Forcing: Next-token Prediction Meets Full-Sequence Diffusion" (NeurIPS 2024)
>
> [3] Karczewski et al. "Diffusion Models as Cartoonists: The Curious Case of High Density Regions" (ICLR 2025)
>
> ---
>
> We thank the Reviewer again for positive feedback and insightful questions regarding potential applications of our proposed framework, and hope that our answers cleared any remaining doubts.

---

### Author Response · Authors · 2025-11-20
**Global Response**

We thank the Reviewers for their feedback and insightful questions, and comments. Below, we address a point raised by multiple reviewers.

### Theoretical guarantees for finding true geodesics

Two Reviewers (zLiS and r6Rp) asked about guarantees for finding true geodesics with our procedure. Below, we address this question in depth.

#### **TL;DR:**
1. Are we guaranteed to find a geodesic - yes, as long as gradient descent converges.
2. Is the geodesic guaranteed to be a globally minimizing curve - not in general, but we can prove it in some simplified cases.

In more detail:

Geodesics and shortest paths are closely related, but there is a subtle distinction between them. A geodesic is a curve with zero acceleration (we only consider the Levi-Civita connection). **Whenever a geodesic is unique, it is necessarily the shortest path**, i.e. the only curve minimizing both the length and the energy [1]. There exist geometries where geodesics are not unique (e.g., both major and minor arc on the sphere are geodesics, but only one of them minimizes the length). However, every geodesic is "locally shortest", which means that we can consider its segments small enough so that geodesics are unique (contained in "normal neighborhoods").

Furthermore, it can be shown [1] that geodesics can be equivalently defined as "critical curves", i.e., curves whose first variation of the energy vanishes. Therefore, since our procedure for finding the geodesic is simply gradient descent of the energy functional w.r.t. the curve's parameters (We have now added the pseudocode - Algorithm 3 in the updated version of the manuscript), **we are guaranteed to find a critical curve, and thus a geodesic** at convergence.

In general, we cannot guarantee that this geodesic is a curve that globally minimizes the length or energy. However, we can prove that this is always the case in two simplified cases:
* when the data distribution is Gaussian, and
* when the data distribution is a uniform distribution over a finite set (i.e., a fixed training set)

> Sketch of proof:

**In the Gaussian case**, we can show that when $q(x_0)$ is Gaussian, then the statistical manifold defined by the posterior distributions $p(x_0|x_t)$ parametrized by noisy points $x_t$ is simply a reparametrized isotropic Gaussian $(\mu, \sigma^2)$ manifold. This manifold has non-positive curvature, which guarantees the uniqueness of geodesics [1].

**In the finite dataset case**, it can be shown that the true posterior $p(x_0|x_t)$ is always a multinomial distribution over the finite training data [2]. Therefore, the resulting statistical manifold is a simplex. This is known to have spherical geometry [3] restricted to the positive orthant, which also guarantees the uniqueness of geodesics.

We appreciate the opportunity to dive deeper into this question and clarify the guarantees for finding the geodesics and their relation to shortest paths. We will include this discussion, along with formal proofs, in the updated version of the manuscript.

---

[1] do Carmo, "Riemannian Geometry". Springer.

[2] Lukoianov et al., "Locality in Image Diffusion Models Emerges from Data Statistics," (NeurIPS 2025).

[3] Arvanitidis et al. "Pulling back information geometry" (AISTATS 2022)

---

### Author Response · Authors · 2025-12-01
**Reviews and discussion summary**

We thank all Reviewers for their time and valuable feedback and Area Chairs, Senior Area Chairs, and Program Chairs for their important work for the community. Below, we summarize the key elements of the reviews and the discussion.

The reviewers identified multiple strengths of the paper, in particular:
* **Soundness:**
  * "Conceptually sound and mathematically motivated" (r6Rp)
  * "Conceptually novel and rigorous" (XMm1)
  * "Theoretically motivated" (zLiS)
* **Empirical Utility:**
  * "Meaningful semantic comparisons"; "operationally useful" (zLiS)
  * "Framework has potential beyond visualization or interpolation tasks" (XMm1)
* **Potential to Inspire Future Research:**
  * "Opens opportunities for future research" (zLiS)
  * "Can lead to generating more research" (r6Rp)
  * "It may have broad impact on the image editing domain" (ixrc)

**Reviewers zLiS and ixrc strongly support the paper (Score: 8)**. Below, we summarize the weaknesses raised by the remaining Reviewers and our responses:

### Reviewer XMm1 (Score: 6)
* **Weakness 1:** Insufficient justification of "memorylessness"
  * **Response:** We elaborated that since $p(x_T|x_0) \approx p(x_T)$, $x_T$ and $x_0$ are effectively independent. Consequently, $p(x_0|x_T)\approx q(x_0)$.
* **Weakness 2:** Lack of runtime comparisons
  * **Response:** We added exact runtimes comparing our method to baselines, confirming that our approach is significantly more efficient than baselines.

### Reviewer r6Rp (Score: 4)
* **Weakness 1:** No pseudocode describing the algorithm
  * **Response:** We have updated the manuscript to include the full algorithm pseudocode.
* **Weakness 2:** Unclear computational cost compared to baselines.
  * **Response:** We added runtime comparisons demonstrating that while our method is naturally more intensive than simple metrics (LPIPS/SSIM), it is faster than comparable geometric/manifold approaches.
* **Weakness 3:** Questioned the metric's correlation with SSIM rather than LPIPS.
  * **Response:** Clarified that the metric quantifies diffusion effort (noise injection). Since small noise primarily affects high frequencies (fine details), the metric naturally aligns closer to structural measures (SSIM) than deep perceptual ones (LPIPS), making it suitable for tasks like super-resolution.

In addition to the above, we addressed the issues raised by the strongly supportive Reviewers and answered all questions.

---

### Meta-Review · Area_Chair_9bQy · 2025-12-19

**Summary:**

The paper's primary strength is its rigorous information-geometric framework, specifically the introduction of a latent spacetime manifold that resolves the degeneracy inherent in standard deterministic pullback metrics. Reviewers consistently praised this conceptual shift as both novel and mathematically sound, noting that equipping the denoising distributions with a Fisher-Rao metric offers a "compelling reframing" of the diffusion latent space. This theoretical depth is complemented by significant practical utility; the resulting Diffusion Edit Distance (DiffED) provides a principled measure of semantic edit cost, while the application to molecular transition path sampling demonstrates versatility beyond standard image tasks. Furthermore, the authors effectively demonstrated that their approach is computationally superior to existing geometric baselines, offering a 5x speedup over comparable Riemannian methods while enabling interpretable, smooth interpolations that respect the intrinsic data geometry.

**Reviewer Concerns:**

The rebuttal successfully addressed the primary technical concerns. Concerns regarding computational efficiency (XMm1, ixrc, r6Rp) were countered with benchmarks showing a 5x speedup over comparable geometric baselines. Theoretical doubts about geodesic uniqueness and the "memorylessness" assumption (zLiS, r6Rp) were resolved with new derivations and uniqueness proofs for specific distributions. Furthermore, reproducibility gaps (r6Rp) were tackled by the inclusion of full pseudocode for geodesic estimation in the revised manuscript.

However, the metric's alignment with structural (SSIM) rather than perceptual (LPIPS) measures remains an inherent characteristic rather than a resolved flaw (r6Rp); the authors clarified this stems from the metric's sensitivity to high-frequency noise. Additionally, despite relative efficiency gains, the method remains orders of magnitude slower than standard heuristics, leaving concerns about its viability for large-scale applications (zLiS) as an acknowledged trade-off.

**Reviewer Scores:**

Reviewer zLiS: The reviewer was already a strong champion of the paper, highlighting the "compelling reframing" of the latent space. Their main concerns regarding computational cost and stability were adequately addressed by the authors' clarifications on runtime and the theoretical guarantees for geodesics.

Reviewer ixrc: This reviewer was also highly positive but requested quantitative metrics to back up the claim that geodesics follow PF-ODE trajectories. The authors provided this exact experiment in the rebuttal (showing negligible LPIPS distance), which would have solidified their confidence.

Reviewer XMm1: This reviewer was positive about the rigor but hesitant due to the lack of runtime comparisons and questions about the "memorylessness" assumption. The authors provided a clear derivation for the latter and a compelling runtime table (showing their method is orders of magnitude faster than MCMC baselines). With these technical hurdles cleared, the reviewer would likely have bumped their score.

Reviewer r6Rp: This reviewer had the most critical concerns: lack of pseudocode, unclear efficiency compared to geometric baselines, and the metric's correlation with SSIM. The authors addressed every point: they added Algorithm 3 & 4 (pseudocode), demonstrated a 5x speedup over the specific baseline mentioned (Azeglio et al.), and provided a sound intuition for the SSIM correlation.

---

### Decision · Program_Chairs · 2026-01-26

Accept (Oral)